# Data Fusion Methods for Indoor Positioning Systems Based on Channel State Information Fingerprinting

**DOI:** 10.3390/s22228720

**Published:** 2022-11-11

**Authors:** Hailu Tesfay Gidey, Xiansheng Guo, Ke Zhong, Lin Li, Yukun Zhang

**Affiliations:** 1Department of Information and Communication Engineering, University of Electronic Science and Technology of China, Chengdu 611731, China; 2Yangtze Delta Region Institute (Quzhou), University of Electronic Science and Technology of China, Quzhou 324000, China

**Keywords:** indoor positioning, CSI-fingerprinting, parking lots, data fusion, knowledge transfer, principal component analysis, CRLB analysis

## Abstract

Indoor signals are susceptible to NLOS propagation effects, multipath effects, and a dynamic environment, posing more challenges than outdoor signals despite decades of advancements in location services. In modern Wi-Fi networks that support both MIMO and OFDM techniques, Channel State Information (CSI) is now used as an enhanced wireless channel metric replacing the Wi-Fi received signal strength (RSS) fingerprinting method. The indoor multipath effects, however, make it less robust and stable. This study proposes a positive knowledge transfer-based heterogeneous data fusion method for representing the different scenarios of temporal variations in CSI-based fingerprint measurements generated in a complex indoor environment targeting indoor parking lots, while reducing the training calibration overhead. Extensive experiments were performed with real-world scenarios of the indoor parking phenomenon. Results revealed that the proposed algorithm proved to be an efficient algorithm with consistent positioning accuracy across all potential variations. In addition to improving indoor parking location accuracy, the proposed algorithm provides computationally robust and efficient location estimates in dynamic environments. A Cramer-Rao lower bound (CRLB) analysis was also used to estimate the lower bound of the parking lot location error variance under various temporal variation scenarios. Based on analytical derivations, we prove that the lower bound of the variance of the location estimator depends on the (i) angle of the base stations, (ii) number of base stations, (iii) distance between the target and the base station, $d_{jr}$ (iv) correlation of the measurements, $\rho_{rj}^{ai}$ and (v) signal propagation parameters $\sigma_{C}$ and γ.

## 1. Introduction

The rapid development of indoor positioning systems (IPS) has been fueled by the emergence of both fifth- and sixth-generation communication systems (5G and 6G) and the internet of things (IoT), as well as the growing commercial interest in location-based services (LBSs). Due to the incredible development of mobile applications, LBSs have gained significant importance in both industrial and commercial applications such as vehicle indoor parking lots [1], indoor navigation [2], self-driving cars [3], security monitoring, and large venue management [4], military use [5], emergency services, tracking, and tourism, and many others [6,7,8]. Modern vehicles rely heavily on GPS to determine their location; however, GPS-based vehicle positioning frequently fails in typical indoor environments especially underground parking lots [9,10]. This could be justified because the indoor environment setting is described as a more complicated scenario than the outdoor setting, owing to (i) the non-line of sight (NLOS) path as a cause of incoherent propagation caused by various barriers along the transceivers; (ii) inherent heterogeneity of signal distributions caused by the dynamic environment in both temporal and spatial variations; and (iii) severe signal attenuation and/or shielding of satellite signals [9,10]. Despite this, a study found that 95% of the time, cars are in a parking lot or an indoor environment [11]. As a result of the challenging nature of the indoor positioning problem (IPP), various radio frequency (RF)-based wireless technologies have been developed to address the demand for higher positioning accuracy while remaining computationally efficient.

To achieve the desired goal of an IPS, two major predictors must be considered: (i) the type of signal features used to establish the fingerprint, such as received signal strength (RSS) [12], CSI [13], time of arrival (TOA) [14], time difference of arrival (TDOA) [15], angle of arrival (AOA) [16], and so on; and (ii) the underlying network technology that generates the signal features. Despite the fact that each technology and signal feature has its own trade-off that limits the scalability of its implementation, over the last two decades, various wireless positioning technologies, including but not limited to cellular networks (GSM) [17], WLAN-Wi-Fi [18], Bluetooth [19], RFID [20], ultra-wideband (UWB) [21], ZigBee [22], inertial navigation systems [23], geomagnetic [24], visible light communication (VLC) [25], etc. have been proposed and investigated. Regardless of the complexity of the indoor environment, the rapidly growing commercial interest in indoor location-based services (ILBSs) still dictates an effective approach for positioning systems that considers both the cost and practicality of their implementation. Vehicle positioning is critical for applications such as navigation, driver assistance and autonomous driving. Several alternative systems for indoor vehicle parking have been developed, including, for example, intelligent parking systems [26] combining RFID and WSN technologies into intelligent parking management systems that use RFID as a unique identification number on the passive RFID card of the driver [27], hybrid systems that use external sensors as well as in-vehicle sensors [28], multiple fisheye surveillance cameras [29] and 3D light detection and ranging (LIDAR) scanners [30]. Additionally, camera and artificial intelligence technologies such as Fuzzy K-means (FCM) and particle swarm optimization (PSO) classification have been used as parameter improvements for parking space detection [31]. Moreover, since the BLE (Bluetooth Low Energy) sensor consumes less power, an intelligent parking system through BLE is possible [32]. 

However, due to the additional infrastructure required for their implementation, most indoor positioning applications (based on Bluetooth, ultra-wideband (UWB), radio frequency identification (RFID), and others) have extra construction costs. Unlike other wireless IPS, which have been criticized for their limited scalability due to the demand for additional hardware devices, RSS-based Wi-Fi fingerprinting is gaining popularity due to its low computational complexity, cost-effectiveness, and ease of implementation and compatibility with existing network infrastructure [33,34,35]. A study [36] proposed an improved Wi-Fi fingerprinting method to replace the behavior of the indoor GPS environment to provide a reliable method of locating vehicles indoors, while also preserving the architecture of the vehicle locating system and facilitating a smooth transition from the outdoor to the indoor, and vice versa. However, due to the complexity of a typical indoor environment, RSS-based fingerprint indoor positioning is susceptible to signal fluctuations, resulting in inefficiency and poor overall positioning performance. The signal fluctuations of the RSS measurements of both instances of training and testing datasets for Wi-Fi APs of AP 1, AP 2, AP 3, AP 4, and AP 5 were reported to have standard deviations (in dB) of 16.8, 15.9, 14.5, 17.9, and 17.1 and 15.63, 15.14, 14.40, 17.92, and 0.00, respectively, as illustrated in Tables 1 and 2 of [37]. This experimental result [37] confirms that the temporal variations in signal distributions have a significant impact on indoor positioning performance-based RSS fingerprints, owing to the effects of multipath, NLOS, and channel conditions such as fading, shadowing, and scattering. Indoor positioning-based RSS fingerprints (FBIP-RSS), on the other hand, have been characterized as having poor spatial resolution and low dimensional feature spaces, which directly degrades indoor positioning accuracy [38]. As a result, the system fails to achieve the desired accurate and robust positioning estimates due to four critical predictors associated with the RSS-based fingerprint that determine the quality of the IPS: (i) high temporal signal fluctuations, (ii) RSS measurements highly susceptible to the effect of a typical indoor environment, (iii) low dimensional feature spaces, and (iv) requirements for large size labeled samples [37,38,39,40,41], which is both costly and labor-intensive.

However, as Wi-Fi network technology has advanced, the Channel State Information (CSI) signal feature can now be extracted from commercial Wi-Fi devices using network interface cards (NICs) that can provide multi-channel subcarrier phase and amplitude information, allowing to better characterize the signal propagation model with the help of multiple-input multiple-output (MIMO) and orthogonal frequency division multiplexing (OFDM) technologies. In comparison to RSS signal features, the CSI-signal feature is distinguished by its ability to depict: (a) fine-grained channel features, (b) higher dimension features, (c) the diversified physical layer of both phase and amplitude information, and (d) more robust and temporal stable features. In line with this, a comparative analysis of CSI and RSSI was performed [42], and the study revealed that the physical characteristics of CSI can significantly reduce the problem of the RSSI, such that (a) multipath effect propagation can be better handled, (b) have strong stability, particularly in a static environment, and are relatively stable to the dynamics of the environment, and (c) reduce radio interference of carrier frequency signals [42]. Thus, given the ability of modern Wi-Fi devices to extract CSI measurements using NICs, as well as the previously mentioned strengths of the CSI signal feature, indoor positioning-based CSI fingerprinting (FBIP-CSI) is attracting significant attention for improving indoor positioning performance.

Nonetheless, due to the complex indoor environment of multipath effects, the CSI-based IPS performance still faces challenges in severe dynamic range and fluctuations among high-dimensional channels [43]. Various CSI fingerprinting-based methods have been proposed and achieved high positioning accuracy in addressing both the impact of the multipath effect as well as signal fluctuations in a dynamic indoor environment [44,45,46]. A novel indoor localization system, FIFS (fine-grained indoor fingerprinting system), using CSI fingerprinting, has been proposed to address signal fluctuations caused by multipath effects in typical indoor environments [47]. During the matching stage, a probabilistic model was applied to accurately map the observed CSI values [47]. However, the assumption of signal distribution requires that the signals be normally distributed to estimate the target’s position by the weighted average of the CSI amplitude values, which is a real challenge due to the indoor multipath effect and dynamic environment. Furthermore, several machine learning (ML) algorithms have been used to mitigate the indoor multipath effect, including support vector machine (SVM) [48], random forest (RF) [49], k-nearest neighbors (KNN) [50,51], and visibility graph (VG) based methods [52]. Moreover, a study [53] proposed a low-overhead indoor positioning transfer learning system based on improved TrAdaBoost to mitigate environmental changes and new scenarios. The proposed method [53] is robust in time and space, with lower site survey overhead while maintaining the same positioning accuracy. Even though the proposed ML algorithms have improved location accuracy, they still face computational complexity challenges due to the CSI’s higher dimension features. Additionally, the proposed ML algorithms are challenged to provide robust location fingerprint estimation, because (i) the fingerprint database must be kept up to date for robust location fingerprint estimation, (ii) the algorithms rely heavily on individual fingerprint parameters, and (iii) a large number of labeled CSI samples are required during model training, which is also very expensive.

The primary goal of this article is to improve CSI-based fingerprint indoor positioning by using data fusion methods to represent temporal signal variations of a location estimator in indoor parking lots. Furthermore, the heterogeneous transfer learning method (HetTL) was used in conjunction with principal component analysis (PCA) to reduce time complexity and ensure cost effectiveness by avoiding unnecessarily high costs associated with extra Wi-Fi access points (Wi-Fi APs) deployment (sources for irrelevant signal features) from the model. This study, on the other hand, uses Cramer-Rao lower bound analysis to estimate the lower bound variance for the estimator data fusion method of CSI-based fingerprint indoor positioning. In this study, the contributions are fourfold:(1)We proposed a data fusion method to represent temporal signal variations by constructing new feature vector spaces based on the most significant predictors and enabling heterogeneous knowledge transfer with the goal of reducing calibration overhead in an indoor parking system.(2)To efficiently detect parking lots, we used the principal component analysis technique to reduce data noise caused by multipath effects, as the channel state information amplitude or fingerprints received from multiple base stations could be mis-matched with the actual target’s fingerprint patterns. In other words, multiple signals arriving at the receiver end from different paths may cause fingerprint duplication and degrades the overall performance of the system. This refers to the possibility of a high-dimensional curse for CSI-based fingerprinting in indoor parking scenarios.(3)The Cramer Rao lower bound (CRLB) analysis was used to estimate the lower bound variance for the estimator of data fusion methods of a vehicle’s indoor parking lot or to measure the unbiasedness of the location estimator for an indoor parking system-based channel state information fingerprinting.(4)We conducted a comparative analysis of the proposed algorithms in terms of the performance of indoor positioning estimation-based channel state information fingerprinting in comparison with the most popular algorithms in the field of machine learning using predictive modeling as a baseline.

The rest of this study is organized as follows: related works are presented in Section 2. Section 3 describes the framework and problem formulation of fingerprint-based indoor positioning with emphasis on data fusion methods, the process of database construction-based CSI-fingerprinting and the system architecture. Evaluation metrics and the CRLB analysis of data fusion methods applied for location estimation are also presented in Section 3. Experimental results and discussions are presented in Section 4. Finally, conclusions are provided in Section 5.

## 2. Related Works

This section provides a brief overview of fingerprint-based methods and data fusion techniques used to address the indoor positioning problem (IPP), system modeling of IP-based CSI fingerprinting, performance evaluation metrics for positioning estimation, and challenges that limit the applications of various signal features used in IPP. Since vehicle positioning is essential in applications such as indoor parking lots, indoor navigation, driver assistance and autonomous driving, accurate information about mobility patterns and vehicle trajectories are essential to improve positioning performance. Generally, three positioning algorithms [54] are applied in IPS: (a) triangulation, (b) the proximity, and (c) scene analysis or fingerprinting. The triangulation method uses the geometric properties of a triangle to estimate the target’s location and comprises two types: angulation and lateration, referring to measuring the angle and distance from multiple grid points (GPs), respectively. One can ask how do these algorithms and signal features estimate the position of a target in relation to the transmitting node or source anchor. Or, what signal features exactly do the algorithms used to estimate the target’s location?

Various signal features were proposed and investigated to address IPP, mainly: time of arrival (TOA/TDOA) [55], angle of arrival (AOA) [56], received signal strengths (RSS) [57,58,59], and channel state information (CSI) [60,61]. In the lateration algorithm, the distance could be acquired indirectly by measuring the received signal strengths indicator (RSSI), time of arrival (TOA), or time difference of arrival (TDOA). Both the signal features of TOA and TDOA are the most accurate techniques, which can filter out multipath effects in an indoor environment situation despite both requiring a line-of-sight (LOS) path along with the transceiver [62,63], which is infeasible in a complex indoor environment. Moreover, they do require high construction costs to be implemented due to their requirements for extra infrastructure investment pertinent to signal directions and need to store precise timing information [64], or these two signal features need to be precisely synchronized [65,66]. Whereas the AOA signal feature does not require time synchronization between measuring units but the AOA or the angulation method demands extra hardware devices pertinent for signal directions [67].

In contrast, the wireless fidelity signal (Wi-Fi: 802.11) has received significant acceptance for IPS both in academia and industry communities [68] mainly for the following reasons: (a) pervasive penetration of Wireless LAN and deployment of Wi-Fi-enabled mobile devices across the globe (cost-effective as it is adopting the existing wireless network infrastructure); (b) the radio wave covers a wide range with a radius about 300 feet or the widespread of its signal over long distances; and (c) it does not require line-of-sight measurement of base stations [55,69] and achieves high applicability in a complex indoor environment. On the other hand, most indoor localization technologies based on Wi-Fi rely on received signal strengths and can be directly implemented using the existing wireless communications infrastructure without any calibrations. The Wi-Fi received signal strength (RSS) measured in decibel milliwatts (dB) is used to find a relationship between transceivers, or measures the accuracy of localization based on the distances between the mobile user and available Wi-Fi access points [57,58,59] through the third method. The so-called scene analysis or fingerprinting comprises two phases: training and testing phases. The RSS fingerprints of the so-called radio map are first collected from each Wi-Fi access point at multiple locations within the defined grid points (GPs) and a predictive model is trained to learn the ‘signal-to-location’ relationship (training phase). The learned model is then applied to infer the location of the target based on the new measurement obtained (online phase) [57,58,59]. Nevertheless, the positioning characteristics-based RSS fingerprinting still has a fundamental problem in accuracy and robustness in IPS, and both temporal and spatial signal fluctuations lead to spontaneous or not robust localization errors. Furthermore, indoor positioning-based Wi-Fi RSS fingerprints have been characterized by low-dimensional feature spaces and a poor spatial resolution, which directly degenerates the indoor positioning performance [70]. In summary, the system fails to achieve the desired accurate and robust positioning estimates due to four critical predictors associated with the RSS-based fingerprint that determines the quality of the IPS: (i) high temporal signal fluctuations, (ii) RSS measurements highly susceptible to the effect of a typical indoor environment, (iii) low-dimensional feature spaces, and (iv) requirements for a large size of labeled samples [37], which is both costly and labor-time-intensive [37,38,39,40,41]. Moreover, RSS is also highly dependent on the used Wi-Fi chipset and how it estimates and reports the RSS value.

The signal feature of the channel state information, however, has emerged as an enhanced wireless channel metric with significant data throughput [42,43] to replace the received signal strength (RSS) for IPS, in which the Wi-Fi networks used MIMO-OFDM techniques whereby data are modulated on multiple channels in different frequencies and simultaneously transmitted among multiple antenna pairs. Apparently, the high dimensional features are possibly produced in WLAN systems due to the MIMO technology integrated into CSI; therefore, an opportunity exists to improve the positioning performance despite facing computational complexity as the main trade-off that needs to be addressed. Ostensibly, the high dimensions of features on their own may not be a positive predictor for the localization process; thus, identifying the most significant predictors is a must; otherwise, some redundant features may inflate or degrade (i.e., cause model overfitting) the system modeling and cause unjustifiable cost for the extra deployment of Wi-Fi access points (Wi-Fi APs). Moreover, some studies based on CSI have demonstrated improved accuracy over RSS for indoor location estimation [42,43], and this can be justified as the CSI reflects the multipath propagation of the signal, to some extent, better than the RSSI.

Towards this end, several CSI-based object detection schemes in WLAN systems have been studied [43,71,72] and shown that the CSI at each subcarrier in OFDM can be used to characterize the target’s behaviors through the two important features of amplitude and phase fluctuations of the targets in a frequency-selective fading channel. Moreover, with the introduction of MIMO technology, high dimensional CSI features are possibly produced in WLAN systems, and it can be considered as an opportunity to improve the positioning performance, although computational complexity is the main trade-off that needs to be addressed. Additionally, a comparative analysis between CSI and RSSI was conducted [42,73]. The study [42,73] revealed that the physical characteristics of CSI can significantly reduce the problem of RSSI such that (a) multipath effect propagation can be better handled (b) owns strong stability especially in a static environment and relatively stable to the dynamics of the environment (c) reduce radio interference of carrier frequency signals [42,73]. Thus, CSI can present different subcarrier amplitude and phase characteristics for different propagation environments. Notably, the overall structural characteristics of CSI remain relatively stable compared with the RSS signal feature; nevertheless, appropriate signal processing technology (SPT) is required. Hitherto, the CSI-based indoor positioning system still has a challenging problem in severe dynamic range and with fluctuation among high-dimensional channels due to indoor multipath effects [43]. Table 1 below presents the notations used in this study.

### System Model

Channel state information (CSI) has emerged as an enhanced wireless channel metric (significantly enhanced data throughput) [74,75] in place of the received signal strength (RSS) for IPS, in which the Wi-Fi networks use the MIMO-OFDM techniques whereby data are modulated on multiple channels (subcarriers) in different frequencies and simultaneously transmitted among multiple antenna pairs (the 802.11 a/g/n standard). The channel response can be extracted from the receivers in the format of CSI, which reveals a set of channel measurements representing the amplitudes and phases of every channel [43,74,75,76]. Additionally, the receiver signal strength can reflect the channel quality of the transmitter and receiver, which can be analyzed from the CSI obtained from the physical layer. CSI also describes the signal propagation process and shows whether the transmitted signal is affected by scattering, attenuation and other factors in the propagation process. In general, the CSI can provide more detailed channel information for a sample than the RSSI. Thus, the received signal power after the multipath channel in OFDM systems can be represented as:(1)Y=HX+ϕ
where Y and X represent the received and transmitted signal vector, respectively, and H and ϕ represent the channel matrix and AGWN (additive Gaussian white noise), respectively, such that $\phi ∼N\left(0,\sigma^{2}I\right)$. Where I is an identity matrix. Thus, the CSI of all subcarriers can be estimated as:(2)$$\hat{H}=\frac{Y}{X}$$
where $\hat{H}$ denotes the channel frequency response (CFR) in the frequency domain. In the narrow band flat fading OFDM channel, the channel matrix H estimated at the receiver represents the physical layer CSI over multiple sub-carriers with the dimension (the format of the received CSI measurements) $n_{T}\times n_{r}\times n_{m}$; and $n_{T}$, $n_{r}$ and $n_{m}$ represent the number of transmitter antennas, receiver antennas and subcarriers for each antenna pair, respectively. We group the subcarriers of the channel state information for each sample along the transceiver antenna pairs as:(3)$$H=\left[H^{1}(f_{1}),H^{1}(f_{2}),\ldots ,H^{1}(f_{m}),H^{2}(f_{1}),H^{2}(f_{2}),\ldots ,H^{k}(f_{m})\right]$$
where $H^{k}(f_{m})=H_{m}$ denotes the $m^{\mathrm{th}}$ subcarrier of the $k^{\mathrm{th}}$ transmitter-receiver pair. Thus, for each location, there are a total L streams for each sample, where $L=n_{t}\times n_{r}\times n_{m}$. And each group of CSI represents the amplitude and phase of an OFDM subcarrier:(4)$$H_{m}=\left|H_{m}\right|e^{j∠H_{m}}$$
where, $\left|H_{m}\right|$ and $∠H_{m}$ represents the amplitude and phase of $m^{\mathrm{th}}$ subcarriers, respectively. In MIMO [77] systems with p transmit antennas and q receive antennas, CSI is a matrix of p×q dimension, which can be expressed as follows:(5)$$H\left(f_{m}\right)=\left[\begin{array}{cccc} h_{11} & h_{12} & \cdots & h_{1q}\\ h_{21} & h_{22} & \cdots & h_{2q}\\ \vdots & \vdots & \ddots & \vdots \\ h_{p1} & h_{p2} & \cdots & h_{pq} \end{array}\right]$$
where $H_{pq}$ exists in the form of a complex number, which represents the amplitude and phase of the subcarrier of the antenna stream. In [78] proposed a fine-grained indoor localization based on CSI data and FILA (Fine Grain Indoor Localization) weights the filtered CSI and normalizes the power to the center frequency in the band as:(6)$$CSI_{eff}=\frac{1}{M}\sum_{m=1}^{M}\frac{f_{m}}{f_{c}}\left(\left|H_{m}\right|\right)$$
where $CSI_{eff}$ is the effective CSI for distance estimation, M and $f_{c}$ are the number of subcarriers and the calculated center frequency, and $\left|H_{m}\right|$ is the amplitude of the filtered CSI on the $m^{\mathrm{th}}$ subcarrier. The propagation distance between the transceiver can be represented by effective channel state information as:(7)$$d=\frac{1}{4\pi }{\left[{\left(\frac{c}{f_{c}\left|CSI_{eff}\right|}\right)}^{2}\times \varphi \right]}^{1/n}$$
where *d* is the estimated distance (in meters (m)) between the transmitter and receiver in an indoor environment setting, c is the radio wave phase velocity (in m/s), $f_{c}$ is the central frequency of CSI (in Hertz or cycles/seconds), n is the path loss attenuation factor (in dB), and φ is the environmental factor (in dB). The environment setting was being conducted in a specified area in controlled fashion, and assumed a constant environmental factor (φ) to estimate the distance between the transceiver. Additionally, the environment factor mainly describes the fixed values (Friis transmission formula) to be assumed depending on the environmental setting as urban, free space, indoor (NLOS/LOS), suburban, etc. The idea behind the environmental factor describes that the targets are exposed or shared same experiences within the experimental setting of the defined indoor parking region as a baseline though practically difficult to ensure it. One can derive that the functional relationship between CSI values and the distance is not direct proportional such that:(8)$$CSI_{eff}=\frac{c}{f_{c}}\sqrt{\frac{\varphi }{{\left(4\pi d\right)}^{n}}}$$

Similarly, the individual path characteristics in a wireless propagation channel are modeled as a temporal linear filter, known as the channel impulse response (CIR) [79]. Given the time invariant channel, the CIR is defined as: (9)$$h\left(\tau \right)=\sum_{i=0}^{N}a_{i}e^{-j\theta_{i}}\delta \left(\tau -\tau_{i}\right)$$
where $a_{i}$; $\theta_{i}$ and $\tau_{i}$ are the amplitude, phase, and time delay of the $i^{\mathrm{th}}$ path of signal propagation, similarly N+1 and δ(τ) are the total number of multipaths and Dirac delta function, respectively. The CIR is characterized as the channel frequency response (CFR) in the frequency domain, and it has been reported [79] in the commercial off-the-shelf WiFi devices that sampled versions of CFRs are revealed to upper layers in the format of CSI. Thus,
(10)$$H\left(f_{m}\right)=\sum_{i=0}^{N}a_{i}e^{-j\left(f_{m}\tau_{i}+\theta_{i}\right)}$$
where $f_{m}$ is the frequency of the $m^{\mathrm{th}}$ subcarrier and $H(f_{m})$ is CSI at the $m^{\mathrm{th}}$ subcarrier, and each CSI depicts the amplitude and phase of a subcarrier as:(11)$$H\left(f_{m}\right)=\left|H\left(f_{m}\right)\right|e^{j∠\theta_{m}}$$
where $\left|H(f_{m})\right|$ is the amplitude and $∠\theta_{m}$ is the phase of each subcarrier, and the received signal gain (in dB) at each subcarrier is proportional to the amplitude of CSI and can be expressed as:(12)$$\hat{H}\left(f_{m}\right)=20\log\left|H\left(f_{m}\right)\right|$$

Thus, the CSI amplitude is a measure of the power of the Wi-Fi link between the transceiver. Using Equation (10), from Euler’s property, $H(f_{m})$ can be written as:(13)$$H(f_{m})=\sum_{i=1}^{N}a_{i}\left[\cos\left(2\pi f_{m}\tau_{i}\right)-j\sin\left(2\pi f_{m}\tau_{i}\right)\right]$$

And the channel state information amplitude for each subcarrier is
(14)$$\left|H(f_{m})\right|=\sqrt{{\left[\sum_{i=1}^{N}a_{i}\cos\left(2\pi f_{m}\tau_{i}\right)\right]}^{2}+{\left[\sum_{i=1}^{N}a_{i}\sin\left(2\pi f_{m}\tau_{i}\right)\right]}^{2}}$$

In the above Equation (9), the total number of paths are N+1. Specifically, N non-line of sight (NLOS) paths and one line of sight (LOS) path, and $a_{0}$ denotes the attenuation amplitude of the LOS path. Since the attenuation of signal strength along the LOS path is mainly caused by path loss and shadowing [80], $a_{0}$ can be expressed as:(15)$$a_{0}=\frac{\sqrt{G_{Tx}G_{Rx}}\lambda }{{\left(4\pi d_{0}\right)}^{n/2}}H_{0},$$
where λ, $G_{Rx}$ and $G_{Tx}$ represent the wavelength of the transmitted signal (in meters), the antenna gains at the receiver (in dB) and transmitter (in dB), respectively. $d_{0}$ denotes the distance of the LOS path, n is the environmental attenuation factor, and $H_{0}$ represents the attenuation of signal amplitude (in dB) caused by shadowing. The NLOS paths originate from radio reflection and refraction. During each reflection or refraction, only partial energy of the signal is transmitted [80], which can be measured by a reflection or refraction coefficient ξ. Therefore, based on Equation (15), with the refraction coefficient, the amplitude of the $m^{\mathrm{th}}$ path $a_{m}$ can be expressed as:(16)$$a_{m}=\frac{\sqrt{G_{Tx}G_{Rx}}\lambda }{{\left(4\pi d_{m}\right)}^{n/2}}H_{m}\xi^{l_{m}},$$
where ξ∈(0,1) is the reflection coefficient and $l_{m}$ is the number of reflections (refractions) along the $m^{\mathrm{th}}$ path and each refraction is assumed to have the same coefficient. The $d_{m}$ represents the distance of the $m^{\mathrm{th}}$ non-line of sight (NLOS) path and $H_{m}$ denotes the attenuation of signal amplitude caused by shadowing along the $m^{\mathrm{th}}$ path. A simplified wireless propagation model is built by integrating the effects of path loss, shadowing, and multi-path based on Equations (9), (15) and (16).

## 3. Problem Formulation and Framework

This section presents an overview of the framework and problem formulation of CSI-based fingerprint IPS with emphasis on: (a) data fusion techniques, (b) process of database construction-based CSI-fingerprinting for both instances of training and testing datasets, (c) the system architecture of CSI-based fingerprinting for IPS, (d) approaches of knowledge transfer learning and the CRLB analysis applied to IPS.

### 3.1. Data Fusion Techniques

It has been noted in various types of research that RSS fingerprinting based on positioning functionality has a fundamental problem in terms of accuracy and robustness in IPS and that both temporal and spatial signal fluctuations cause spontaneous or inestimable localization errors, regardless of the affordability and ease of implementation. A survey conducted in [81] stated that the fusion of multiple measurements from different sensors has been becoming crucial to improve the positioning performance. It is also clearly noted that the CSI can measure different information to yield a better location estimate than the RSSI [43,82]. In [81], we have also discussed that positioning or tracking based on a single measurement could aggravate the tracking and/or positioning performance. Moreover, hybrid methods were also proposed to enhance indoor positioning performance including the hybrid-based positioning system of different localization applications such as the combinations of Bluetooth, Wi-Fi, UWB and ZigBee [8], Wi-Fi with visual light positioning (VLP) [83], Wi-Fi and Bluetooth low energy (BLE) beacons [84], and others. By analyzing the limitations of signal strength values (RSSI) fingerprint locations, geometric locations, and inertial navigation locations, an indoor data fusion method based on an adaptive unscented Kalman filter (UKF) was proposed [85]. The algorithm uses a six-position error calibration method and Kalman filter to compensate for the MEMS-SINS data and establishes the correlation between location data and RSSI/geomagnetic data based on the feature sorting vector fingerprint matching method, which leads to improved data stability and indoor location accuracy [85]. In this study, the scope of our work is limited to a positioning system based on the various temporal variations collected over different periods of time based on CSI fingerprints. The real measurements of CSI were considered for our analysis. We claim that the data fusion technique as illustrated in Figure 1 below is an effective way to further improve the accuracy and robustness of indoor-based positioning given that the different temporal signal variations are aggregated to account and measure their signal differences, which could produce the net effect of their positioning performance. Figure 1 presents the proposed framework of the CSI-based fingerprint data fusion technique for IPS.

### 3.2. System Architecture and Database Fingerprint Construction

We have noted that positioning characteristics-based RSS fingerprinting still has a fundamental problem in accuracy and robustness in IPS, and both temporal and spatial signal fluctuations cause spontaneous or inestimable localization errors. It has been shown that the CSI at each subcarrier in OFDM can be used to characterize the target’s behaviors through the two important features of amplitude and phase fluctuations of the targets in a frequency-selective fading channel. Additionally, due to the MIMO technology integrated into CSI, high-dimensional features are possibly produced in WLAN systems and can be considered as opportunity to improve the positioning performance, although computational complexity is the main trade-off needed to be addressed. Of course, the high dimensions of features by its own may not be a positive predictor for the localization process; thus, identifying the most significant predictors is a must otherwise some redundant features may degrade or inflate (i.e., cause for model overfitting) the system modeling. In line with this, a comparative analysis between CSI and RSSI was conducted and the study revealed that the physical characteristics of CSI can significantly reduce the problem of RSSI, such that (a) multipath effect propagation can be better handled (b) strong stability is provided, especially in a static environment, and relatively stable to the dynamics of the indoor environment (c) reduce radio interference of carrier frequency signals. 

In this study, as depicted in Figure 2 below we adopted the second method, the so-called scene analysis to construct a database of fingerprints that comprises two main phases: training and testing phases. The CSI fingerprints are first collected from each base station (BSs) located at four different locations (there were four BSs in total) within the defined reference points (RPs) and partitioned into two parts as a training set and testing set for different purposes. Suppose $\left\{c_{rj}^{ai}(t_{d}),a\in [1,n_{a}],i\in [1,n_{c}]\right\}$ denote the $i^{\mathrm{th}}$ CSI amplitude value at the $r^{\mathrm{th}}$ RP of the $j^{\mathrm{th}}$ BS from $a^{\mathrm{th}}$ receiver antenna (Rx). The $t_{d}$ refers to the time when the data were collected, specifically measured in number of days. $n_{c}$ and $n_{a}$ are the number of CSI measurements at each RP and number of antennas of a BS, respectively. Let $C_{k}^{q}\left(t\right)$ represent the aggregated values of $c_{rj}^{ai}(t_{d})$, which are collected from all RPs of the $j^{\mathrm{th}}$ BS. Let’s consider that the CSI measurements were collected on different days of two months of September and October, 2020 represented as $C_{k}^{q}\left(t_{1}\right)$ and $C_{k}^{q}\left(t_{2}\right)$, and $\rho_{rj}^{ai}$ denote the correlation coefficient between the two vectors as given in Equation (31). Now, we can formulate the problem mathematically such that the general fingerprint-CSI based positioning of the indoor environment scenario is divided into R reference points (RPs). Each RP represents a target’s location and is indexed with a label r,(r=0,1,…,R−1). There were b detectable base stations (j=1,2,…,b) in total. Thus, the $i^{\mathrm{th}}$ CSI amplitude value at the $r^{\mathrm{th}}$ RP of the $j^{\mathrm{th}}$ BS from $a^{\mathrm{th}}$ antenna in a particular day of $t_{d}$ forms a vector and the fingerprint database can be represented as a multi-dimensional matrix such that $C_{rj}^{ai}(t_{d})$:(17)$$C_{rj}^{ai}(t_{d}|\left\{t_{d}=d_{1},\ldots ,d_{n}\}\right)=\left[\begin{array}{c} \begin{array}{ccc} c_{01}^{11} & \cdots & c_{01}^{1n}\\ \vdots & \ddots & \vdots \\ c_{(R-1)1}^{11} & \cdots & c_{(R-1)1}^{1n} \end{array}\\ \begin{array}{ccc} c_{01}^{21} & \cdots & c_{01}^{2n}\\ \vdots & \ddots & \vdots \\ c_{(R-1)1}^{21} & \cdots & c_{(R-1)1}^{2n} \end{array}\\ \begin{array}{ccc} c_{(R-1)1}^{n_{a}1} & \cdots & c_{(R-1)1}^{n_{a}n}\\ \vdots & \ddots & \vdots \\ c_{(R-1)1}^{n_{a}1} & \cdots & c_{(R-1)1}^{n_{a}n} \end{array} \end{array}\right]$$

The fingerprint database is described explicitly as: $C_{rj}^{ai}(t_{d})=\left\{C_{rj}^{ai}(t_{d});\left(x_{r},y_{r}\right)\right\};r=0,1,\ldots ,R-1;i=1,2,\ldots ,n\&j=1,2,\ldots ,b$, and $\left(x_{r},y_{r}\right)$ is the corresponding coordinate to the associated location of the CSI signature fingerprint. The training set constituted about 80 percent of the total instances, and the remaining are allotted to the testing dataset. To this end, the predictive models were trained to characterize the ‘signal-to-location’ relationship given the training instances and their corresponding labels of the location were stored in the database fingerprints (offline phase). This process was repeated for all RPs to characterize or store the signal signature of a reference point with their corresponding location. The learned models were then applied to infer the location of the target’s location based on the testing data points or measurements by mapping into the highest likelihood similarity of signal-signature stored during the training phase time. The target instances would be the testing dataset and denoted as $\left\{C_{rj}^{ai}\right\};i=1,2,\ldots ,n_{t}\&j=1,2,\ldots ,b$  and the training dataset representing the source domain can be represented as: $C_{rj}^{ai}=\left\{C_{rj}^{ai};\left(x_{r},y_{r}\right)\right\};r=0,1,\ldots ,R-1;i=1,2,\ldots ,n_{s}\&j=1,2,\ldots ,b_{s}$ called the labeled source data. $n_{t}$ and $n_{s}$  represent the numbers of measurements for the target and source data instances, respectively. Figure 2 below demonstrates the proposed system architecture of the CSI-based fingerprint for indoor positioning system.

### 3.3. Principal Component Analysis (PCA)

With the advancement of Wi-Fi network technology, the CSI signal feature can now be extracted from commercial Wi-Fi devices using network interface cards (NICs) that can provide multi-channel subcarrier phase and amplitude information, allowing to better characterize the signal propagation model with the help of MIMO-OFDM technologies. Thus, CSI can present different subcarrier amplitude and phase characteristics for different propagation environments and the overall structure characteristics of CSI remain relatively stable compared with the RSS signal feature, though appropriate signal processing technology (SPT) is required. Nevertheless, the CSI-based IPS still has a challenging problem in severe dynamic range and fluctuations among high-dimensional channels mainly due to indoor multipath effects and temporal variations. This could be justified because the spread of the CSI in both the amplitude and phase values measured at a particular reference point is so dynamic in nature. Additionally, the inherent heterogeneity of the environment resulting from multipath effects and time differences of when measurements were taken have a significant effect on the distribution of the CSI values received at each reference point. Thus, there is no guarantee that the signal variations be represented by a single value for a target’s position even with the same device. To address these issues, we proposed a data fusion technique for reducing channel fluctuations and improving target positioning-based heterogeneous knowledge transfer by creating a new feature vector based on the most significant predictors. Furthermore, we used the principal component analysis (PCA) technique to reduce noise and irrelevant or insignificant features, because the CSI amplitude or fingerprints received from multiple base station antennas could be mismatched with the actual target’s fingerprint patterns due to fingerprint duplication. Moreover, with the use of the PCA, we addressed the high-dimensional curse for the CSI-based fingerprint method. The data preprocessing of PCA was used to reduce computational complexity and ensure cost effectiveness by avoiding unnecessarily high costs associated with extra Wi-Fi access points (Wi-Fi APs) deployment (sources for irrelevant signal features) from the model.

Recall that the ith CSI amplitude values received at the rth RP of the jth BS from ath antenna in a particular day of td forms a vector, and the fingerprint database collected on different days can be represented as a multi-dimensional matrix. Each vector of the CSI amplitude measurements are assumed to be independent and identically distributed such that the joint probability density function with p dimension of random vectors of the CSI amplitude values observed at different temporal variations can be defined as:(18)f(Crjai(s),μ)=∏i=1n{1(2π)p/2|∑|1/2e−12(ℂ−μ)′∑−1(ℂ−μ)}
where Crjai(s)=ℂ and μ represents all possible measurements that the CSI amplitude values could take and the mean, respectively, and this joint normal density is denoted as Crjai(s)∼N(μ,∑). Similarly, the covariance of CSI amplitude measurements from several antennas of a base station can be determined as:(19)cov(Sz,Sq)=∑(Sz−S¯z)(Sq−S¯q)n−1

The variances of the measurements Sz and Sq can be computed as:(20)sd(Sq)=∑i=1n(ℂqi−ℂ¯q)2nq−1&sd(Sz)=∑i=1n(ℂzi−ℂ¯z)2nz−1
where the sample means of the CSI amplitude for the random vectors of Z and Q can be given as: (21)ℂ¯z=1nz∑i=1nzCrjai(sz)&ℂ¯q=1nq∑i=1nqCrjai(sq)

Equivalently, we can write the above equation in (19) as: (22)∑i=1n(ℂi−ℂ¯i)(ℂi−ℂ¯i)T=ℂℂT

To this end, the PCA algorithm comprises four major steps:(a)Standardize each CSI measurement as: C′rjai(sz)=Crjai(sz)−1nz∑i=1nzCrjai(sz)(b)Calculate the covariance matrix of the CSI sample measurements: ∑i=1n(ℂi−ℂi)(ℂi−ℂ¯i)T=ℂℂT(c)Obtain Eigen value decomposition of covariance matrix(d)Obtain projection matrix

Algorithm 1 is a pseudo code used to derive the new fingerprints of features-based CSI using both data fusion techniques and principal component analysis.

**Algorithm 1** Construction of the new dimension of CSI fingerprint feature vectors-based data fusion and PCA1. **Input:** Multiple Sources Fingerprints Crjai(str)={Crjai(s1);Crjai(s2),…,Crjai(sl);(xr,yr)};a=[1,na];j=1,2,…,b2. **Input:** CSI testing datasetCrjai(sts)={Crjai(s1);Crjai(s2),…,Crjai(sl);(?,?)};a=[1,na];j=1,2,…,b
3. **Output:** Refined CSI Fingerprint C′rjai(s′)={C′rjai(s′1);C′rjai(s′2),…,C′rjai(s′l);(x′r,y′r)};a=[1,na];j=1,2,…,b4. **for**
r=0:R−1 do5.    **for**
j=0:b−1 do6.     **for**
$a=0:n_{a}-1$ do**7.      for**i=0:n−1 do8.     fuse the data as $C_{rj}^{ai}(s^{f})=\left\{{C^{\prime }}_{rj}^{ai}({s^{\prime }}_{1});{C^{\prime }}_{rj}^{ai}({s^{\prime }}_{2}),\ldots ,{C^{\prime }}_{rj}^{ai}({s^{\prime }}_{l});\left({x^{\prime }}_{r},{y^{\prime }}_{r}\right)\right\};a=[1,n_{a}];j=1,2,\ldots ,b$
9.             Standardize each CSI measurement using (21)10.           Calculate the covariance matrix of the CSI measurements as in (19) or (22) 11.           Compute the eigenvectors and eigenvalues of the new vector feature 12.          Create feature vectors $g_{1},g_{2},\ldots ,g_{n^{\prime }}$ corresponding to the largest $C_{rj}^{ai}(s^{f})$ eigenvalues.13.          Obtain projection matrix: $G^{*}=\left(g_{1},g_{2},\ldots ,g_{n^{\prime }}\right)$
14.     **end for**15.    **end for**16.   **end for**17. **end for**18. return $\left\{{C^{\prime }}_{rj}^{ai}(s_{tr}^{f},)(x_{r}^{\prime },y_{r}^{\prime })\},\{{C^{\prime }}_{rj}^{ai}(s_{ts}^{f}),\left({\hat{x}}_{r}^{\prime },{\hat{y}}_{r}^{\prime }\right)\right\}$

### 3.4. Heterogeneous Knowledge Transfer

In this study, the CSI fingerprints were first collected from each antenna of a base station (BS) located at four different locations (there were four BSs in total) within the defined reference points (RPs) and partitioned into two parts as t training set and testing set for different purposes. Along with this, we have noted that there are two major challenges that need to be considered in heterogenous transfer learning: (i) the CSI amplitude and phase received at a reference point from multiple antennas of a base station assumed to be independent such that the RF signals transmitted from different BSs are transmitted independently and do not interfere with each other. In practice, however, the CSI amplitude of a grid point can be duplicated, possibly by multiple BSs, and this causes the matching patterns to interfere with the actual target’s fingerprint due to random noise or indoor multipath effect scenarios and ultimately have a negative effect on positioning performance. (ii) Although channel state information could help us provide more feature spaces, the higher dimensional curse must be handled. Thus, we proposed the data fusion technique to minimize the temporal CSI amplitude fluctuations by considering different measurements over several days that could have different patterns due to the inherent environmental heterogeneity and multipath effects. Additionally, potentially duplicated CSI fingerprints and the dependence of CSI measurements on different BS antennas were managed through principal component analysis.

Thus, the objective function was minimized over the new feature spaces such that the most significant features, independent features, and related source knowledge could be leveraged to the target domain. The Minkowski is a generalized distance metric between two vectors and given as:(23)$$\left\|r\right\|=\sum_{i^{\prime }=1}^{n}\left\|C_{rj}^{ai^{\prime }}(s_{tr}^{f})-{\hat{C}}_{rj}^{ai^{\prime }}(s_{ts}^{f})\right\|$$

${\left\|r\right\|}_{1}$ and ${\left\|r\right\|}_{2}$ denotes Manhattan and Euclidean distances, respectively. Similarly, $C_{rj}^{ai^{\prime }}(s_{tr}^{f})$ and ${\hat{C}}_{rj}^{ai^{\prime }}(s_{ts}^{f})$ represent the fused CSI amplitude values of instances of training and testing datasets respectively. The transfer coefficients $\left(\omega_{i^{\prime }j^{\prime }}\right)$ constraint is to minimize the amplitude measurements of fluctuations between the instances of both the training and testing dataset. The objective function’s equality constraint would assign higher weights to the most related source instances and lower weights to the least related source instances. The new feature vectors were used to minimize the variations of the amplitude values of CSI based fingerprints and the transfer coefficients can be estimated as:(24)$$\begin{array}{l} \min\sum_{i^{\prime }=1}^{n^{\prime }}\sum_{j^{\prime }=1}^{n^{\prime }}\omega_{i^{\prime }j^{\prime }}{\left\|C_{rj^{\prime }}^{ai^{\prime }}(s_{tr}^{f})-{\hat{C}}_{rj^{\prime }}^{ai^{\prime }}(s_{ts}^{f})\right\|}_{2}^{2}\\ s.t.\sum_{i^{\prime }=1}^{n^{\prime }}e^{-\omega_{mi}}=1,i^{\prime }=1,2,\ldots ,{n^{\prime }}_{t} \end{array}$$

We have used the Lagrangian multiplier method to solve the constrained optimization problem of Equation (24), and we assumed the location estimate at the ${\left(t-1\right)}^{\mathrm{th}}$ iteration $\omega_{{}_{i^{\prime }j^{\prime }}}^{\left(t-1\right)}$ is obtained (mapped into 2D), and we need to estimate the location of the actual target at the $t^{\mathrm{th}}$ iteration denoted by $\omega_{{}_{i^{\prime }j^{\prime }}}^{\left(t\right)}$. CSI fingerprints are collected from each Wi-Fi access point at multiple locations within the reference points (RPs) defined during the training phase. A predictive model is trained to characterize the signal-to-location learning relationship. During the testing phase, the learned model is then used to predict the target’s location based on the new received CSI measurement. One can rewrite Equation (24) using the Lagrangian multiplier method as:(25)$$L\left(\omega_{ij},\lambda \right)=\sum_{i^{\prime }=1}^{n^{\prime }}\sum_{j^{\prime }=1}^{n^{\prime }}\omega_{i^{\prime }j^{\prime }}{\left\|C_{rj}^{ai^{\prime }}(s_{tr}^{f})-{\hat{C}}_{rj^{\prime }}^{ai^{\prime }}(s_{ts}^{f})\right\|}_{2}^{2}+\lambda \left(\sum_{i^{\prime }=1}^{n^{\prime }}e^{-\omega_{i^{\prime }j^{\prime }}}-1\right)=0$$
where λ is the Lagrangian multiplier. By letting the partial derivative of the Lagrangian with respect to $\omega_{i^{\prime }j^{\prime }}$ and λ be zeros, we obtain:(26)$$\frac{\partial }{\partial \omega_{i^{\prime }j^{\prime }}}L\left(\alpha_{j^{\prime }},\lambda \right)=\left\{ \begin{array}{l} {\left\|C_{rj^{\prime }}^{ai^{\prime }}(s_{tr}^{f})-{\hat{C}}_{rj^{\prime }}^{ai^{\prime }}(s_{ts}^{f})\right\|}_{2}^{2}-\lambda e^{-\omega_{1j^{\prime }}}=0\\ {\left\|C_{rj^{\prime }}^{ai^{\prime }}(s_{tr}^{f})-{\hat{C}}_{rj^{\prime }}^{ai^{\prime }}(s_{ts}^{f})\right\|}_{2}^{2}-\lambda e^{-\omega_{2j^{\prime }}}=0\\ \prime \\ \prime \\ {\left\|C_{rj^{\prime }}^{ai^{\prime }}(s_{tr}^{f})-{\hat{C}}_{rj^{\prime }}^{ai^{\prime }}(s_{ts}^{f})\right\|}_{2}^{2}-\lambda e^{-\omega_{n_{s}{}_{j^{\prime }}}}=0\\ \sum_{i=1}^{n}e^{-\omega_{i^{\prime }j^{\prime }}}-1=0 \end{array}\right.$$

By adding up the first n terms in Equation (26), we can obtain:(27)$$\lambda ={\sum_{i^{\prime }=1}^{n}\left\|C_{rj^{\prime }}^{ai^{\prime }}(s_{tr}^{f})-{\hat{C}}_{rj^{\prime }}^{ai^{\prime }}(s_{ts}^{f})\right\|}_{2}^{2}$$

And substituting Equation (27) into Equation (26) gives the estimated transfer coefficients as:(28)$$\omega_{i^{\prime }j^{\prime }}=-\ln\left(\frac{{\left\|C_{rj^{\prime }}^{ai^{\prime }}(s_{tr}^{f})-{\hat{C}}_{rj^{\prime }}^{ai^{\prime }}(s_{ts}^{f})\right\|}_{2}^{2}}{\sum_{i^{\prime }=1}^{n}{\left\|C_{rj^{\prime }}^{ai^{\prime }}(s_{tr}^{f})-{\hat{C}}_{rj^{\prime }}^{ai^{\prime }}(s_{ts}^{f})\right\|}_{2}^{2}}\right)$$

The pseudo code for positioning using the heterogenous knowledge transfer-based data fusion technique is provided in Algorithm 2.

**Algorithm 2** The proposed heterogenous knowledge transfer-based CSI-fingerprint indoor positioning system1. **Input:** Refined Sources Fingerprint C′rjai(sd)|tr={C′rjai(s1);C′rjai(s2),…,C′rjai(sl);(x′r,y′r)};a=[1,na];j=1,2,…,b
2. **Input:** Refined CSI testing datasetC′rjai(sd)ts={C′rjai(s1t);C′rjai(s2t),…,C′rjai(slt);(?,?)};a=[1,na];j=1,2,…,b
3. **Output:** Fused refined CSI fingerprints C′rjai(strf),C′rjai(stsf), position estimates, transfer coefficients4. **for**
r=0:R−1 do5.  **for**
j=0:b−1 do6.   **for**
$a=0:n_{a}-1$ do7.  **for**
i=0:n−1 do
8. **repeat**
9.   **Step 1:**10.     Fuse the CSI amplitude fingerprints from all the sources with temporal variations as in Algorithm 1
11.   **Step 2:**12.      Obtain projection matrix: G∗=(g1,g2,…,gc′)13.   **Step 3:**14.      Compute ωi′j′f by using Equations (24)–(28)15.      **end for**16.     **end for**17.    **end for**18. **end for**19. Train a classifier from C′rjai(strf)={C′rjai(strf);(xr′f,yr′f)} while considering weights of source domains ωi′j′f
20. Estimate target’s location (xr′f,yr′f) on {C′rjai(stsf)} by applying the trained classifier f({{C′rjai(strf)};(xr′f,yr′f)},ωi′j′f) 
21. **return** {C′rjai(strf);(xr′f,yr′f)tr},{C′rjai(stsf);(x^r′f,y^r′f)ts},ωi′j′f


### 3.5. Evaluation Metrics for Positioning

This section presents the metrics applied to evaluate the positioning performance, and we compared the performance of the proposed algorithms against different machine learning algorithms taken as baselines and validated through extensive real-life experimentations. The dataset were collected over several days with potential temporal signal fluctuations based on CSI-fingerprinting of the real measurements. In this study, the root mean square error was used to evaluate the positioning performance of the proposed algorithm, and it is defined as:(29)$$RMSE={\left[\frac{1}{n_{t}}\sum_{i^{\prime }=1}^{n_{t}}\left[{\left(x_{i^{\prime }}-{\hat{x}}_{i^{\prime }}\right)}^{2}+{\left(y_{i^{\prime }}-{\hat{y}}_{i^{\prime }}\right)}^{2}\right]\right]}^{1/2}$$
where ${\left[{\hat{x}}_{i^{\prime }},{\hat{y}}_{i^{\prime }}\right]}^{T}$ and ${\left[x_{i^{\prime }},y_{i^{\prime }}\right]}^{T}$ are the predicted location estimate and actual location of the target, respectively. And $n_{t}$ is the total number of samples to be located in the target domain.

### 3.6. Cramer-Rao Lower Bound (CRLB) Analysis for IP Performance-Based CSI-Fingerprinting

This section presents the analysis of CRLB for location estimation-based channel state information-measurements used to evaluate the performance of indoor positioning system and estimates a lower limit for the variance of any unbiased estimator of an unknown parameter. Additionally, the CRLB is suitable for stationary gaussian parameter estimation [86,87,88,89]. The CSI-based fingerprint database is denoted as: Crjai(td)={Crjai(td);(xr,yr)};r=0,1,…,R−1;i=1,2,…,n&j=1,2,…,b and (xr,yr) is the corresponding coordinate to the associated location of the CSI signature fingerprint. Thus, we proposed the data fusion technique to represent the temporal CSI amplitude fluctuations by considering different measurements over several days that could have different patterns due to the inherent environmental heterogeneity and multipath effect. Suppose the ith CSI amplitude values received at the rth RP of the jth BS from ath antenna in a particular day of td forms a vector. Thus, the fingerprint database collected on different days can be represented as a multi-dimensional matrix and each vector of the CSI amplitude measurements are assumed to be independent and identically distributed such that the probability density function of a random vector of the CSI amplitude values observed in particular days or time measurement td can be defined as:(30)f(Crjai(s),μ)=12πσe−12(ℂ−μσ)2
where Crjai(s)=ℂ and μ represents all possible measurements that the CSI amplitude values could take and the mean, respectively, and this joint normal density is denoted as Crjai(s)∼N(μ,σ2). Similarly, the correlation of CSI amplitude measurements from several antennas of a base state can be determined as:(31)corr(ℂx,ℂy)=cov(ℂx,ℂy)sd(ℂx)×sd(ℂy)

In line with this, the distributions of the CSI values measured at a particular reference point are so dynamic in nature due to the inherent heterogeneity of the environment, and time differences in measurements being taken have a significant effect on the distribution of the CSI values received at each RP. As a result, there is no guarantee that the signal fluctuations will be represented by a single value for a specific position even with the same device. The CRLB used the CSI real measurements to analyze the lower bound of the location estimation error, and it is significantly important to characterize the properties of this lower bound to evaluate the impact of different parameters on the accuracy of a target’s localization. Moreover, the CRLB analysis can also provide important system design suggestions by revealing error trends with the indoor localization system deployment. To estimate the lower bound of the position error variance, first assume that the position of the $j^{\mathrm{th}}$ base station and the unknown location of the target are denoted as $L_{j}={\left(x_{j},y_{j}\right)}^{T}$ and ${\hat{L}}_{j}={\left({\hat{x}}_{j},{\hat{y}}_{j}\right)}^{T}$, respectively, then the distance between the $j^{\mathrm{th}}$ base station and the target can be defined as $r_{0;j}^{2}=\left\{\left(L_{j}-\hat{L}\right){\left(L_{j}-\hat{L}\right)}^{T}\right\}$. It has been stated that the CSI fingerprint measurements follow a normal distribution with mean zero and variance $\sigma^{2}$. Thus, we adopted the assumption of normality and the covariance of the estimator $\hat{L}$, which is n×2, vector can be defined as:(32)$$E\left\{\left(L_{j}-\hat{L}\right){\left(L_{j}-\hat{L}\right)}^{T}\right\}=\left[\begin{array}{cc} E{\left(x_{j}-{\hat{x}}_{j}\right)}^{2} & E\left(x_{j}-{\hat{x}}_{j}\right)E\left(y_{j}-{\hat{y}}_{j}\right)\\ E\left(y_{j}-{\hat{y}}_{j}\right)E\left(x_{j}-{\hat{x}}_{j}\right) & E{\left(y_{j}-{\hat{y}}_{j}\right)}^{2} \end{array}\right]$$

The diagonal elements of (32) denote the mean squared errors, and the off-diagonal elements are the covariances between different parameters. Additionally, consider that $L_{j}$ is the unknown deterministic parameter, which is to be estimated from n independent observations (CSI fingerprint-based measurements collected over different periods of time) of $C_{rj}^{ai}(s)$, each from a distribution according to some probability density function $f\left(C_{rj}^{ai}(s);L_{j}\right)$ . Thus, the CRLB is defined as the variance of any unbiased estimator ${\hat{L}}_{j}$ of $L_{j}$ bounded by the reciprocal of the Fisher information matrix $I\left(L_{j}\right)$ such that: (33)$$\sigma^{2}\left({\hat{L}}_{j}\right)\geq {\left[I\left(L_{j}\right)\right]}^{-1}$$

And, the I(Lj) is given as:(34)I(Lj)=[Ixx(Lj)Ixy(Lj)Iyx(Lj)Iyy(Lj)]=nEL[(∂l(ℂ;Lj)∂Lj)2]
where l(Crjai(s);Lj)=log(f(Crjai(s);Lj)) is the natural logarithm of the likelihood function for a single sample Crjai(s) and EL denotes the expected value with respect to the density function of ℂ,f(Crjai(s);Lj). But, if l(Crjai(s);Lj) is twice differentiable and holds certain regularity conditions, then the Fisher information can also be defined as: (35)I(Lj)=−nEL(∂2l(ℂ;Lj)∂Lj2)

By definition, the CRLB of the unbiased estimator ${\hat{L}}_{i}$ can be calculated as:(36)$$CRLB=\frac{I_{xx_{i}}\left(L_{j}\right)+I_{yy_{i}}\left(L_{j}\right)}{\left|I(L_{j})\right|}$$

One can describe the CRLB as just the inverse of FIM and rewrite Equation (19) as:(37)$$Cov(L_{j})\geq {\left[I(L_{j})\right]}^{-1}=\frac{1}{\left|I(L_{j})\right|}\left[\begin{array}{cc} I_{yy}(L_{j}) & -I_{yx}(L_{j})\\ -I_{xy}(L_{j}) & I_{xx}(L_{j}) \end{array}\right]$$

Thus, the variance of the unbiased estimator for $L_{j}$ is given by: (38)$$\mathrm{var}\left({\hat{L}}_{j}\right)\geq E{\left(x_{j}-{\hat{x}}_{j}\right)}^{2}+E{\left(y_{j}-{\hat{y}}_{j}\right)}^{2}$$

Now, from the above Equations (33) and (37), one can observe the relationship between the variance of the unbiased estimator ${\hat{L}}_{j}$ and FIM, which basically satisfies the CRLB condition such that:(39)$$v\left({\hat{L}}_{j}\right)=E{\left(x_{j}-{\hat{x}}_{j}\right)}^{2}+E{\left(y_{j}-{\hat{y}}_{j}\right)}^{2}\geq tr\left\{{\left[I\left(L_{j}\right)\right]}^{-1}\right\}$$
(40)$$tr{\left[I\left(L_{j}\right)\right]}^{-1}=\frac{I_{xx_{i}}\left(L_{j}\right)+I_{yy_{i}}\left(L_{j}\right)}{\left|I(L_{j})\right|}$$

One can observe from Equation (38), the MSE of ${\hat{x}}_{j}$ and ${\hat{y}}_{j}$ can be given as:(41)$$MSE\left({\hat{x}}_{j}\right)=E{\left(x_{j}-{\hat{x}}_{j}\right)}^{2}\geq \frac{I_{y_{j}y_{j}}\left(L_{j}\right)}{\left|I\left(L_{j}\right)\right|}$$
(42)$$MSE\left({\hat{y}}_{j}\right)=E{\left(y_{j}-{\hat{y}}_{j}\right)}^{2}\geq \frac{I_{x_{j}x_{j}}\left(L_{j}\right)}{\left|I\left(L_{j}\right)\right|}$$
where $tr\left\{{\left[I\left(L_{j}\right)\right]}^{-1}\right\}$ denotes the trace of the inverse FIM and $\left|I(L_{j})\right|$ represents the determinant of FIM.

In this study, the multiple measurements of CSI-based fingerprints collected on different days were used as different sources to estimate the lower bound of the error variance of the target’s location using the application of CRLB. We considered that the CSI amplitudes collected on various days follow a multivariate normal distribution. This is consistent with the assumption that different signal features including the well-known features such as RSS, CSI, TOA, and AOA follow a normal distribution [89,90]. Recall that the ith CSI amplitude values received at the rth RP of the jth BS from ath antenna in a particular day of td forms a vector. The fingerprint database collected on different days can be represented as a multi-dimensional matrix and each vector of the CSI amplitude measurements of a day are assumed to be independent and identically distributed such that the joint probability density function with p dimension of random vectors of the CSI amplitude values observed on different temporal variations can be defined as in Equation (18):(43)f(Crjai(s),μ)=∏i=1n{1(2π)p/2|∑|1/2e−12(ℂ−μ)′∑−1(ℂ−μ)}
(44)∑=(σ12ρ12σ1σ2⋯ρ1sσ1σsρ21σ2σ1σ22⋯ρ2sσ2σs⋮⋮⋱⋮ρs1σsσ1ρs2σsσ2⋯σs2)where *s* is the number of CSI source feature vectors, σm2 is the variance of the sth source feature of the CSI measurements, ρis is the correlation coefficient between sources of measurements collected on different days, and ∑ represents the multidimensional covariance of different sources of CSI measurements. Let djr denote the distance between the unknown location of the target and the base station, and given as (xj−x^j)2+(yj−y^j)2. One also can establish the geometric relationship between the two coordinates of the target’s location and base station. Consider the following Figure 3 to demonstrate the relationship based on angle. 

Thus, one can denote $u_{jr}^{2}=\frac{\sum_{j=1}^{n}{\left(x_{j}-{\hat{x}}_{j}\right)}^{2}}{d_{jr}^{4}}$, $v_{jr}^{2}=\frac{\sum_{j=1}^{n}{\left(y_{j}-{\hat{y}}_{j}\right)}^{2}}{d_{jr}^{4}}$, and $u_{jr}v_{jr}=\frac{\sum_{j=1}^{n}\left(x_{j}-{\hat{x}}_{j}\right)\left(y_{j}-{\hat{y}}_{j}\right)}{d_{jr}^{4}}$. The CRLB of multiple CSI fingerprint-based measurements for localization can also be described as in [89]:(45)$$CRLB\left(C_{rj}^{ai}(s_{i:1\to n})\right)=\frac{1}{G}\left[\frac{u_{jr}^{2}+v_{jr}^{2}}{u_{jr}^{2}\times v_{jr}^{2}-{\left(u_{jr}v_{jr}\right)}^{2}}\right]$$
where:(46)$$G=\left[\alpha_{1},\alpha_{2},\ldots ,\alpha_{n}\right]\sum^{-1}\left[\begin{array}{l} \alpha_{1}\\ \vdots \\ \alpha_{n} \end{array}\right]$$

Scenario 1: Consider eight temporal variations of measurements of month 1 as different sources due to their distribution follow different patterns: Thus, we consider eight different sources for evaluating the positioning performance
(47)$$G=\left[\alpha_{1},\alpha_{2},\alpha_{3},\alpha_{4},\alpha_{5},\alpha_{6},\alpha_{7},\alpha_{8}\right]{\left(\begin{array}{cccc} \sigma_{1}^{2} & \rho_{12}\sigma_{1}\sigma_{2} & \cdots & \rho_{18}\sigma_{1}\sigma_{8}\\ \rho_{21}\sigma_{2}\sigma_{1} & \sigma_{2}^{2} & \cdots & \rho_{28}\sigma_{2}\sigma_{8}\\ \vdots & \vdots & \ddots & \vdots \\ \rho_{81}\sigma_{8}\sigma_{1} & \rho_{82}\sigma_{8}\sigma_{2} & \cdots & \sigma_{8}^{2} \end{array}\right)}^{-1}\left[\begin{array}{l} \alpha_{1}\\ \alpha_{2}\\ \alpha_{3}\\ \alpha_{4}\\ \alpha_{5}\\ \alpha_{6}\\ \alpha_{7}\\ \alpha_{8} \end{array}\right]$$

And,
(48)$$CRLB\left(C_{rj}^{ai}(s_{i:1\to 8})\right)=\frac{1}{G}\left[\frac{u_{jr}^{2}+v_{jr}^{2}}{u_{jr}^{2}\times v_{jr}^{2}-{\left(u_{jr}v_{jr}\right)}^{2}}\right]$$

Scenario 2: Consider five temporal variations of measurements of month 2 as different sources due to their distribution follow different patterns: Thus, we consider five different sources for evaluating the positioning performance
(49)$$G=\left[\alpha_{1},\alpha_{2},\alpha_{3},\alpha_{4},\alpha_{5}\right]{\left(\begin{array}{cccc} \sigma_{1}^{2} & \rho_{12}\sigma_{1}\sigma_{2} & \cdots & \rho_{15}\sigma_{1}\sigma_{5}\\ \rho_{21}\sigma_{2}\sigma_{1} & \sigma_{2}^{2} & \cdots & \rho_{25}\sigma_{2}\sigma_{5}\\ \vdots & \vdots & \ddots & \vdots \\ \rho_{51}\sigma_{5}\sigma_{1} & \rho_{52}\sigma_{5}\sigma_{2} & \cdots & \sigma_{5}^{2} \end{array}\right)}^{-1}\left[\begin{array}{l} \alpha_{1}\\ \alpha_{2}\\ \alpha_{3}\\ \alpha_{4}\\ \alpha_{5} \end{array}\right]$$

And,
(50)$$CRLB\left(C_{rj}^{ai}(s_{i:1\to 5})\right)=\frac{1}{G}\left[\frac{u_{jr}^{2}+v_{jr}^{2}}{u_{jr}^{2}\times v_{jr}^{2}-{\left(u_{jr}v_{jr}\right)}^{2}}\right]$$

Scenario 3: Consider the aggregate temporal variations of measurements of various sources of both months into two different sources and calculate the CRLB of those fused different sources for evaluating the positioning performance purpose:(51)$$G=\left[\alpha_{1}^{f},\alpha_{2}^{f}\right]{\left(\begin{array}{cc} \sigma_{1f}^{2} & \rho_{12}\sigma_{1f}\sigma_{2f}\\ \rho_{21}\sigma_{1f}\sigma_{2f} & \sigma_{2f}^{2} \end{array}\right)}^{-1}\left[\begin{array}{l} \alpha_{1}^{f}\\ \alpha_{2}^{f} \end{array}\right]$$

And
(52)$$CRLB\left(C_{rj}^{ai}(s_{1}^{f},s_{2}^{f})\right)=\frac{\sigma_{1f}^{2}\sigma_{2f}^{2}-\rho_{12}\sigma_{1f}^{2}\sigma_{2f}^{2}}{\alpha_{1f}^{2}\sigma_{2f}^{2}+\alpha_{2f}^{2}\sigma_{1f}^{2}-\rho_{12}\sigma_{1f}\sigma_{2f}\alpha_{1f}\alpha_{2f}}\left[\frac{u_{jr}^{2}+v_{jr}^{2}}{u_{jr}^{2}\times v_{jr}^{2}-{\left(u_{jr}v_{jr}\right)}^{2}}\right]$$

Now, to estimate the lower bound of the position error variance, first consider that the position of the $j^{\mathrm{th}}$ base station and unknown location of the target are denoted as $L_{j}={\left(x_{j},y_{j}\right)}^{T}$ and ${\hat{L}}_{j}={\left({\hat{x}}_{j},{\hat{y}}_{j}\right)}^{T}$ respectively; then, the distance between the $j^{\mathrm{th}}$ base station and $r^{\mathrm{th}}$ reference point can be defined as $r_{0;j}^{2}=\left\{\left(L_{j}-\hat{L}\right){\left(L_{j}-\hat{L}\right)}^{T}\right\}$, and for simplicity purposes let’s use $r_{jr}^{2}$ instead of $r_{0;j}^{2}$. Similarly, the distance between the $j^{\mathrm{th}}$ base station and target point can be represented as $r_{ji}^{2}$. Researchers have found that the CSI fingerprint measurements follow a normal distribution with mean zero and variance $\sigma^{2}$. Besides, $n_{b}$ and γ represent the number of base stations and path loss attenuation factor, respectively, and $\sigma_{C}$ is the variance of flat fading, and multipath follows normal distribution. In [91] have proposed a model for the effective vector of CSI from $j^{\mathrm{th}}$ AP measured at RP and given as
(53)$$\ln CSI_{eff,r}^{2}=V_{rj}=\ln\left(\frac{c^{2}\delta }{f_{c}^{2}{\left(4\pi r_{rj}\right)}^{\gamma }}\right)+\varepsilon_{r}$$
where δ is an environment factor, γ is the path loss attenuation factor, c is the radio velocity, and $\varepsilon_{r}$ is a measurement noise and follows normal distribution $\varepsilon_{r}∼N\left(0,\sigma_{r}^{2}\right).$ The same device in an unknown location $L_{j}=\left(x_{j},y_{j}\right)$ measures a CSI value of $V_{ij}$ from the same AP given as:(54)$$\ln CSI_{eff,i}^{2}=V_{ij}=\ln\left(\frac{c^{2}\delta }{f_{c}^{2}{\left(4\pi r_{ij}\right)}^{\gamma }}\right)+\varepsilon_{i}$$

Similarly, $\varepsilon_{i}∼N\left(0,\sigma_{i}^{2}\right).$ When the fingerprint is used for localization, the AP locate at an unknown location and we utilize the fingerprint to estimate the coordinate of $L_{j}$. $H_{k}$ is defined as the difference between $H_{rj}$ and $H_{ij}$ given by:(55)$$H_{ij}-H_{rj}\left[\mathrm{dB}\right]=\ln{\left(\frac{r_{ij}}{r_{rj}}\right)}^{\gamma }+\varepsilon_{d}$$

Thus, the probability density function of the Hk given the location f(Hk|Lj) or the pdf of the estimated location-based CSI-fingerprint is given by:(56)f(Hk(sk)|Lj)=∏i=1n{12πσCe−12σC2(Hk(sk)−ln(rijrrj)γ)2}where Crjai(sm)=Hk(sk)=Hij(sk)−Hrj(sk) denotes the difference in the effective vector of the CSI measurement representing the ith target at the rth RP of the jth base station. The CRLB-based CSI-fingerprint based on Equation (45) can also be given as:(57)CRLB(ℂ)=1αC[ujr2+vjr2ujr2×vjr2−(ujrvjr)2]
where αC=(γ2σC)2. Consider the special case of the above equation in (1) and the joint probability density function with p=2 dimensions of random vectors of the CSI amplitude values observed on different temporal variations can be defined as:(58)f(Crjai(S1f,S2f))=12πσs1σs21−ρ2exp{−12(1−ρ2)[(ℂ1−μs1σs1)2−2ρ(ℂ1−μs1σs1)(ℂ2−μs2σs2)+(ℂ2−μs2σs2)2]}
where ρ denotes the correlation coefficient between the two fused CSI fingerprints collected in month 1 and month 2. Let μ=[μs1,μs2]T and Σ=(σs12ρ12σs1σs2ρ21σs1σs2σs22). Similarly, based on the above Equation (57), the probability density function of the estimated location-based CSI-fingerprint for p=2 dimensions can be given as:(59)f(Crjai(S1f,S2f))=12πσs1σs21−ρs1s22exp{−12(1−ρs1s22)[A]}
(60)A=(H1(s1)−ln(rjrrir)γ)2σs12−2ρs1s2(H1(s1)−ln(rjrrir)γσs12)(H2(s2)−ln(rjrrir)γσs22)+(H2(s2)−ln(rjrrir)γ)2σs22

Thus, one can rewrite Equation (57) for the CRLB based CSI-fingerprint measurements in relation to the angle as follows:(61)CRLB(ℂ)=1αC[∑r=1R(cosβjrdjr)2+∑r=1R(sinβjrdjr)2∑r=1R(cosβjrdjr)2×∑r=1R(sinβjrdjr)2−(∑r=1R(sinβjrcosβjrdjr2))2]

The CRLB of the fused data from different sources can be calculated based on the concept given in the above Equation (52), and the CRLB or lower bound error of the location for the two fused sources of CSI-based fingerprint measurements collected in two separate months of several days can be given as:(62)CRLB(ℂ)=1−ρs1s22αs1+αs2−2ρs1s2(αs1αs2)1/2[∑r=1R(cosβjrdjr)2+∑r=1R(sinβjrdjr)2∑r=1R(cosβjrdjr)2×∑r=1R(sinβjrdjr)2−(∑r=1R(sinβjrcosβjrdjr2))2]

In conclusion, the above derivation revealed that the lower bound of the variance of the location estimator depends on (a) the angle of the base stations, (b) number of base stations, (c) distance between the target and base station, $d_{jr}$, (d) correlation of features, ρ, and (e) signal propagation parameters, $\sigma_{C}$ and γ. Moreover, experimental results have shown that the number of antennas of a base station could affect the lower bound location estimation error by generating a higher dimension of features, unless the most significant predictors are selected; otherwise, the accuracy of positioning performance could be degraded due the dimensionality curse. This analytical derivation also revealed that the fused data have shown the hybrid effect of the temporal signal variations resulting due to time differences in the measurements being taken.

## 4. Experimental Results and Discussion

This section presents a number of real-world experiments that were carried out on various occasions, or on a daily basis to measure the temporal signal fluctuations using the fingerprints of the channel state information (interested in variations among the measurement days), and we assess our proposed algorithms as they were applied to these CSI real measurements of various distributions. First, experimental conditions, datasets, and an analysis of the algorithms’ overall performance were presented.

### 4.1. Experimental Settings

(1)
**Datasets**


The experiments were carried out at Huawei on an area of 75 m^2^, which contained 225 and 110 reference points (RPs), respectively, all of which were evenly dispersed (≥0.5 m) from each other as shown in Figure 4. A first survey was conducted in September 2020, and a second survey was conducted in October 2020. For the first month, eight measurements were taken on eight different days. In contrast, five different measurements were taken on five separate days in the second month. To create the entire CSI fingerprint database, four base stations and one transmitter were available to collect channel state information from a location server. The number of reference points collected per day are different in both months. The total number of reference points in each month also differ. Since the measurements under study are naturally unbalanced, our analysis must account for possible discrepancies caused by the imbalance of data. Details of the system description used in the study are provided in Table 2 below. The layout and environmental settings of the real-life experimental scenarios generated for the various datasets are depicted in Figure 4 and Figure 5 below.

### 4.2. Distribution of the Temporal Variations of CSI Amplitude Measurements

Table 3 depicts the distribution of channel state information per label, specifically the number of measurements collected from each reference point (RP), and demonstrates that the label distribution is unbalanced for all datasets collected for the eight different days of September 2020, noted as month 1. In contrast, the number of measurements collected from each reference point (RP) are all equal, indicating that the label distribution for all datasets collected during the five separate days of month 2, October 2020, is balanced. Feature-scaling techniques were used in this study to avoid the dominance of the larger occurrence of labels within the cluster, which would otherwise cause the larger features to dominate the others within the cluster and negatively affects modeling performance. However, for the month 2 data collection, an equal number of CSI fingerprint measurements were recorded for all RPs, and thus the reference point distribution was considered balanced for all datasets collected in October 2020. In general, principal component analysis is recommended to reduce the effect of the dimensionality curse on large datasets with higher dimensions of features where computational complexity is significant. For the use of principal component analysis, features must be standardized. According to Table 3, the total number of RPs collected during the first month is 225, and the total number of labels collected during the second month is 110. The CSI fingerprint measurements features collected during the survey time provide complete information that assisted in determining target positioning, including CSI real measurements, CSI imaginary measurements, latitude, longitude, coordinate systems, time of arrival, angle of arrival, and other relevant data for all scenarios.

Figure 6 depicts the distribution of the principal components and their corresponding reference points of CSI-based fingerprint measurements collected on 16 September 2020, noted as month 1 and day 1 (d1M1). Despite a minor difference caused by temporal signal fluctuations generated within the environment, the CSI-based measurements of both training and testing datasets received from various antennas of a base station at a reference location appear to follow a specific distribution. The first and second principal components of both instances from the training and testing datasets provide the highest proportion of variance explainability to the positioning model and allow one to visualize the effect of variations in the target’s location using the most significant predictors. The first and second principal components could explain the total variance of the system model as a linear combination of the first and second principal components. However, as illustrated in Figure 6c,d, the two principal components of the last principal, which actually constitute about 4% of the total variations in the predictive model for both instances of the training and testing dataset, appear to follow different distributions or the testing dataset appears to fail to represent the training distribution. While principal component analysis is important for minimizing computational complexity, removing significant features has a critical impact on positioning accuracy and negatively impacts positioning performance. A “base model,” according to [38], consists of only two principal components to represent the variational distributions of the target’s prediction, accounting for approximately 16% of the total variance explainability of the model, though [38] achieved 56%. Even though the base model has a lower variance explainability ratio of 16% versus 56%, this finding is consistent with the findings in [38] that lower feature space dimensions can improve computational cost and model simplicity. This clearly demonstrates that the base model was unable to fit our problem with the desired goal, necessitating a larger number of principal components. As a result, we proposed in this study the number of principal components that could account for 95% of the explained variance ratio of system performance in general, but these dimensions of principal components must also verify if the desired system modeling with the expected positioning accuracy can be maintained. Similarly, Figure 7 depicts the distribution of the principal components and their corresponding reference points of CSI-based fingerprint measurements collected on 9 October 2020, abbreviated as month 2 and day 1 (d1M2).

The distribution of the principal components and their corresponding reference points of both training and testing instances of the fused CSI-based fingerprint of real measurements collected over several days in October 2020 is depicted in Figure 8. Unlike the previous daily-based distributions, which were limited to a few specific reference points, the fused data distribution of the entire dataset considered days of month 2 would give us a complete picture of the positioning system. The fused data, on the other hand, represents the entire distribution of the CSI fingerprint measurements over the total defined grid points where positioning performance can be established given the possibility of signal fluctuations from all reference points where targets can be located. The average amplitude and signal fluctuation values are more representative than individual day measurements. Moreover, the total number of reference points collected during the survey period of October 2020 across all days of consideration was 110. Furthermore, despite a slight variation due to temporal signal fluctuations generated within the environment, the fused CSI-based measurements of both training and testing datasets received from multiple antennas of a base station at a reference point appear to obey a specific distribution. One can also see that the first and second principal components of both instances from the training and testing datasets account for the greatest proportion of the variance explainability of the positioning model. This could explain the total variance of the system model as a linear combination of those most significant features dedicated to positioning performance. However, the two final principal components shown in Figure 8c,d, which account for about 4% of the total variations of the predictive model for both fused instances of the training and testing dataset, appear to follow different distributions. In other words, the fused testing dataset appears to fail to represent the fused training dataset distribution.

Table 4 and Table 5 demonstrate the effect of various feature space possibilities on the amount of variance ratio explainability of the predictive models for indoor parking lots or target localization accuracy, considering the nature of distributions of fused instances of both training and testing data points separately collected over several days in September and October 2020, respectively. The model’s total variations of the fused data could be explained by 16 principal components of both the training and testing datasets. Similarly, the 90% explained variance ratio required 14 principal components to fit the system modeling or indoor parking lot scenario in both fused training and testing phases, and this difference in principal components was found to be insignificant. The first and second principal component distributions account for approximately 23% of the total model variations that could be accounted for by the fused data of both the training and testing datasets, separately. This explains why the two distributions of the base model appear the same, why the CSI fingerprint measurements received at a reference point from different base stations appear to be independent, and why the distribution of each RP seems difficult to characterize with the base model, which consists of only two main components accounting for approximately 23% of the total variance of the predictive model. Although limited and significant predictors could represent the vehicle’s indoor parking system more effectively and cheaply, a more considerable number of dimensions of feature spaces or principal components are required to fit the model. In other words, the base model, which accounts for approximately 23% of the variance ratio’s explainability, does not fully reflect the model’s variation and thus does not fit the indoor parking system scenario. This is consistent with the finding that the less variance explained ratio by a principal component, the less likely the algorithms are to characterize the ‘CSI amplitude-location’ relationship, and thus fail to estimate the target position or indoor parking lots [92,93].

### 4.3. Comparative Analysis of Methods

This section presents extensive real-life experiments conducted on various occasions or daily to measure the temporal signal fluctuations observed for indoor parking lots-based CSI fingerprint measurements. We compared our proposed algorithms applied to these CSI-based real measurements of different distributions to popular machine learning algorithms used in prediction tasks. Table 6 depicts the application of several machine learning algorithms to real-world scenarios of indoor parking lots based on CSI fingerprint measurements collected on different days of month 1, specifically September 2020. In this study, we are particularly interested in modeling vehicles’ indoor parking lots based on CSI-based fingerprints of amplitude values from various datasets collected over several days. Data preprocessing was also performed to understand the details of the datasets and gain insights for possible hypothesis or claim generation useful to address the indoor parking problem, and our analysis took those factors into account. Extensive experimentation was performed in this regard on real-life scenarios of the indoor parking phenomenon, and the experimental results revealed that the proposed algorithm was found to be an efficient algorithm with a consistent score of positioning accuracy across the potential dynamics of temporal variations while the average execution testing time was higher than that of the other algorithms, which is the trade-off to be penalized. Furthermore, unlike the support vector machine and neural network algorithms, which consume enormous computational complexity, the proposed algorithm not only improved indoor parking localization accuracy but was also found to be a computationally robust positioning estimator. Results also revealed that the training computational complexity was significantly higher than the testing computational time for all algorithms. However, the support vector machine and neural network algorithms demanded an unusually large amount of computational complexity time during the training and testing phases, following the proposed algorithm.

In contrast, results show that both decision tree (DT) and random forest (RF) classifiers performed better in separate datasets of both months (Table 6, Table 7, Table 8 and Table 9). However, their positioning accuracy was found to be lower than the proposed algorithm for the fused dataset, which represented the entire temporal signal variations of the indoor environment and was used to make our final decision (Table 10). Furthermore, both the decision tree and random forest models were found to be highly inconsistent with positioning performance across all potential temporal dynamics. This explains why both the DT and RF classifiers were highly susceptible to signal fluctuations caused by the dynamic indoor environment and multipath effects, as these models rely on the randomness of the selected feature. However, the positioning accuracy of these classifiers for all the separate datasets from both months have improved significantly after transfer learning was applied, but not for the fused dataset. This implies that the estimator of positioning based on this DT and RF classifiers is highly inconsistent and unstable in the dynamic indoor environment. As shown in Table 9, these classifiers’ positioning accuracy was surprisingly reduced after the PCA technique was used for the fused dataset to reduce computational complexity. Thus, overall metrics such as unbiasedness, efficiency, consistency, and average execution testing time have demonstrated that the proposed algorithm is the best algorithm to model indoor parking lots based on CSI fingerprinting of the fused data. 

Table 7 illustrates the use of heterogeneous knowledge transfer applied to real-life scenarios of indoor parking lots based on the CSI fingerprint measurements collected on separate days of month 1, September 2020. The experimental results show that the proposed algorithm was found to be an efficient algorithm with a consistent score of positioning accuracy across the potential dynamics of temporal variations that were being observed in the indoor parking problem domain. Additionally, the proposed algorithm has not only managed to improve indoor parking positioning performance through heterogeneous knowledge transfer but also, computationally cost-wise, was relatively efficient, unlike the support vector machine and neural network algorithms, which involve huge computational complexity. Moreover, we observed that the computational complexity was much higher during the application of heterogeneous knowledge transfer for all algorithms in general, and this is the penalty cost to obtain better positioning performance. However, we exceptionally observed that too much computational complexity was demanded by the support vector machine for both the training and testing phases, followed by the neural network algorithm.

Table 8 below demonstrates the use of several machine learning algorithms applied to real life scenarios of indoor parking lots based on the CSI fingerprint measurements collected on separate five days of month 2, October 2020 and the fused data, respectively. Along with this, extensive experimentation applied to real life scenarios of indoor parking phenomenon were conducted, and the experimental results revealed that the proposed algorithm was found to be an efficient algorithm with a consistent score of positioning accuracy across the possible dynamics of temporal variations observed during October 2020 for five separate days. The proposed algorithm has not only managed to improve indoor parking localization accuracy but also computationally cost-wise was also efficient, unlike the support vector machine and neural network algorithms that involve huge computational complexity. Moreover, as noted above, we also observed that the training computational complexity was much higher for all algorithms than testing time in general. We have also exceptionally observed too much computational complexity time was demanded by the support vector machine for training and testing phases, followed by the neural network algorithm.

Table 9 shows the application of heterogeneous knowledge transfer to real-life scenarios of indoor parking lots based on CSI-fingerprint measurements collected on separate five days of month 2, on October 2020, and the experimental results show that the proposed algorithm was found to be efficient with a consistent score of positioning accuracy across the possible dynamics of temporal variations that were observed in indoor parking probing. In contrast to the support vector machine and neural network algorithms, heterogeneous knowledge transfer not only improved indoor parking positioning performance but was also relatively efficient in terms of computational cost. Furthermore, the survey, which was conducted over five days in October 2020, revealed that the computational complexity was much higher during the application of heterogeneous knowledge transfer for all algorithms in general, and this is the trade-off we pay to achieve better positioning performance. Similarly, we observed that the support vector machine demanded an unusually large amount of computational time during the training and testing phases, followed by the neural network algorithm.

On the other side, Table 10 presents models vehicle’s indoor parking lots of CSI-based fingerprints of the fused data-based PCA method. The indoor parking positioning performance significantly improved after transfer learning method was applied for both the fused data and the separate datasets. The positioning performance for the separate datasets of both months has significantly reduced after principal component was used to minimize computational cost. This revealed that the use of PCA method or considering 95% of the total variations of the model could not represent the entire dynamics of the environment but knowledge transfer could leverage from the training instances to enhance target positioning. However, the positioning performance for the fused data of month 2, October 2020 has maintained equal performance as that of the performances achieved in separate dataset while significant improvement was observed in computational cost. Moreover, heterogeneous knowledge transfer of the fused data based PCA not only improved indoor parking positioning performance but was also significantly efficient in terms of computational cost as can be seen in Figure 8 and Figure 9.

Moreover, the following Table 11 shows the parameters specification being used by each algorithm in the study.

### 4.4. Computational Complexity Analysis

This section compares the computational complexity of our proposed algorithm, which was applied to the fused CSI-based real measurements of different distributions collected in separate days of two months, to popular machine learning algorithms used in prediction tasks to represent the temporal signal variations observed for indoor parking lots. The algorithms in this study were executing on a laptop computer equipped with an AMD Ryzen 3 3200U CPU (2.60 GHz) and 16 GB of RAM. The complexity of an algorithm is primarily determined by two scenarios of complexity: time and space. We used both the functional analysis of Big O and average elapsed execution testing time to compare the time complexity of the algorithms. The Big O functional analysis of time complexity provided us with the overall worst-case computational cost analysis, and all of the algorithms we used have an average of worst-case time complexity of $O(n^{2}).$ However, the proposed algorithm’s average elapsed execution testing time is much higher than that of the other algorithms that we used. Figure 9 illustrates the comparative analysis of computational testing time of the fused data before and after PCA was applied. We noted that the fused data represents the entire distribution of the CSI fingerprint measurements over the total defined grid points where positioning performance could be established given the possibility of signal fluctuations that could come from all reference points where targets could be located. Accordingly, the computational testing time for the proposed algorithm is higher followed by the neural network (MLP) and SVC. However, after PCA was applied, the computational testing time was significantly reduced or improved while maintaining the same order.

Figure 10 depicts the performance analysis of classifiers applied to indoor parking lots based on fused data of CSI-fingerprints. We have applied three scenarios to analyze the positioning performance as a) the fused data b) the fused data based PCA and c) the fused data in conjunction with PCA and transfer learning. Thus, the proposed algorithm (scenario c) has the best positioning performance with minimum root mean square error applied to indoor parking lots based on the fused data of the CSI-fingerprint measurements.

## 5. Conclusions

In this study, we considered various scenarios of temporal variations that generated the CSI-based fingerprinting measurements applied to indoor environment settings aimed at vehicles’ indoor parking lots. Along with this, extensive real-life experiments were conducted at Huawei company in a different time with an area of 75 m^2^ constituting 225 and 110 reference points (RPs) in total. The data were collected over separate dates on September and October 2020, respectively. Each RP was equidistant (>=0.5 m) from the next reference point. The number of measurements considered from each antenna of a base station was unequal in size. Similarly, the number of labels was not totally balanced. Thus, our analysis used the feature scaling technique to avoid possible discrepancies created due to the imbalance of data. To this end, we proposed a heterogeneous data fusion method based on positive knowledge transfer to represent the different temporal variation scenarios of CSI-based fingerprint measurements generated in a complex indoor environment targeting indoor parking lots, while reducing training calibration overheads. Extensive experiments were carried out with real-world scenarios of the indoor parking phenomenon. Experimental results revealed that the proposed algorithm proved to be an efficient algorithm with consistent positioning accuracy across all potential variations. The proposed algorithm not only improves indoor parking location accuracy, but it is also a computationally robust and efficient estimator for a dynamic indoor environment unlike the decision tree and random forest algorithms, which are significantly affected by the temporal signal fluctuations.

The results also show that the training computational complexity for all algorithms was much higher than the overall testing time, and that the proposed algorithm required higher computational complexity for the training and testing phases, followed by support vector machine and neural network algorithms. However, an interesting finding was observed for the vehicle’s indoor parking lots of CSI-based fingerprints of the fused data-based PCA method. The indoor parking positioning performance significantly improved after the transfer learning method was applied to both the fused data and separate datasets. Nevertheless, the positioning performance for the separate datasets of both months was significantly reduced after the principal component method was used to minimize computational cost. This revealed that the use of the PCA method, or considering 95% of the total variations of the model, could not represent the entire dynamics of the environment, but knowledge transfer could be leveraged from the training instances to enhance target positioning. In contrast, the positioning performance for the fused data of month 2, October 2020 maintained equal performance as that achieved in separate dataset, while significant improvement was also observed in computational cost. Moreover, heterogeneous knowledge transfer of the fused data-based PCA not only improved indoor parking positioning performance, but was also very efficient in terms of computational cost as depicted in Figure 8 and Figure 9. This exactly coincides with our claim that the fused data is significantly important for representing the signal fluctuations-based CSI-fingerprints in a dynamic environment, typically in an underground parking plot.

The CRLB analysis technique was also applied to estimate the lower bound of the position error variance aimed at indoor parking lots. Similarly, different scenarios of temporal variations were also considered for CRLB analysis of CSI-based fingerprint measurements applied to indoor environment settings, such as vehicles’ indoor parking lots. Thus, the analytical derivation of the CRLB analysis revealed that the lower bound of the variance of the location estimator depends on (a) the angle of the base stations, (b) the number of base stations, (c) the distance between the target and base station, $d_{jr}$ (d) correlation of features, $\rho_{rj}^{ai}$, and (e) the signal propagation parameters $\sigma_{C}$ and γ. Moreover, experimental results have shown that the number of antennas of a base station could affect the lower bound of the variance of the location estimator by generating a higher dimension of features unless the most significant predictors are selected; otherwise, the accuracy of positioning performance could be degraded due to the dimensionality curse. This analytical derivation also revealed that the fused data have shown the hybrid effect of the temporal signal variations that could come through time differences when measurements were being taken. 

The database consists of values of different signal features as clearly mentioned on Section 4.2. The CSI fingerprint measurements collected during the survey provide information to aid in target positioning, including real CSI measurements, imaginary CSI measurements, latitude, longitude, coordinate systems, time of arrival, angle of arrival and other relevant data for all scenarios. Thus, the original database is a vast one consisting of various information. But, we extract data from a database that suits our research goal. Following are the recommendations we forwarded as future research directions:(1)Even though we limited our scope to CSI-amplitude information based on our objective, the phase information of CSI can also be used as fingerprints to location.(2)Fusion of various signal measurements also could result in a robust and efficient estimator for parking lots, although advanced signal processing technology is required in real life to minimize computational cost.(3)Correlation feature extraction of the various signal metrics can also be considered in addressing signal fluctuations resulted due to the dynamic environment.(4)Effective data preprocessing approaches are highly recommended in improving the positioning performance.(5)Handling the data dimensionality curse could also improve the computational complexity. Thus, various approaches of data reduction techniques are highly important and worth further investigations.

## Figures and Tables

**Figure 1 sensors-22-08720-f001:**
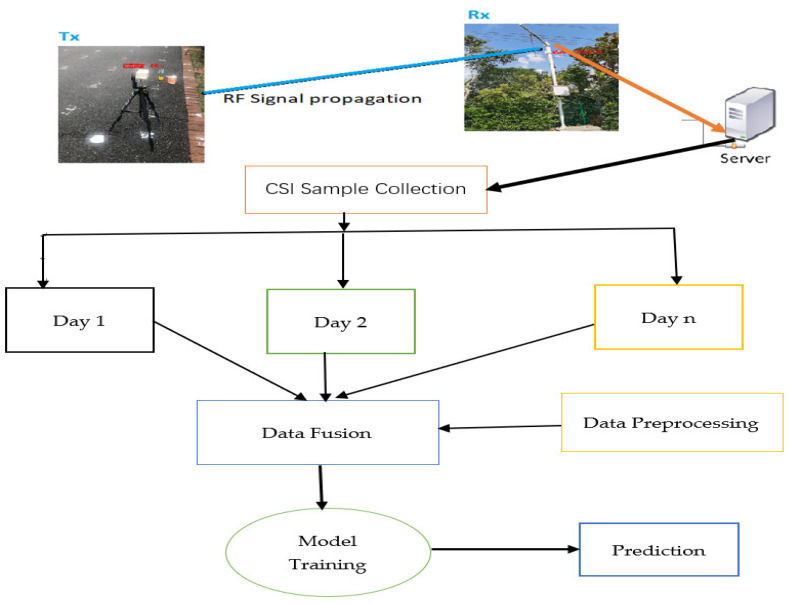
The proposed framework of the CSI-based Fingerprint Data Fusion technique for an indoor positioning system.

**Figure 2 sensors-22-08720-f002:**
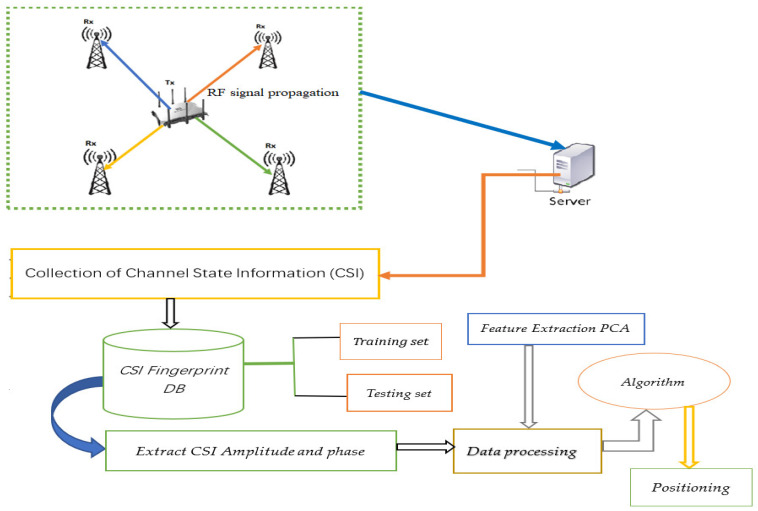
Proposed system architecture of CSI-based Fingerprint Database Construction for Indoor Positioning System.

**Figure 3 sensors-22-08720-f003:**
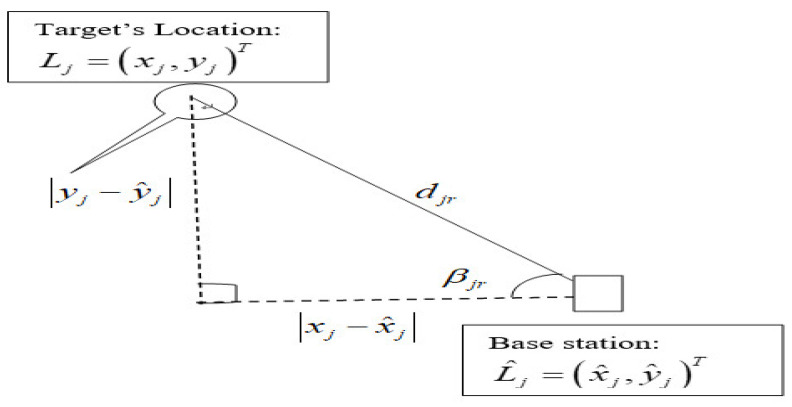
Geometric representation of the distance between the Target’s position and Base station.

**Figure 4 sensors-22-08720-f004:**
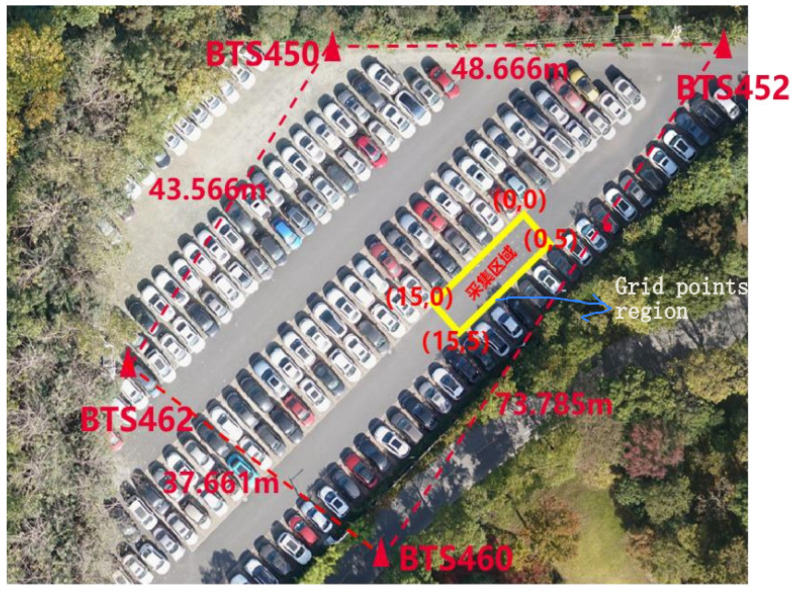
Experimental layout conducted at Huawei company for indoor parking-based CSI fingerprints.

**Figure 5 sensors-22-08720-f005:**
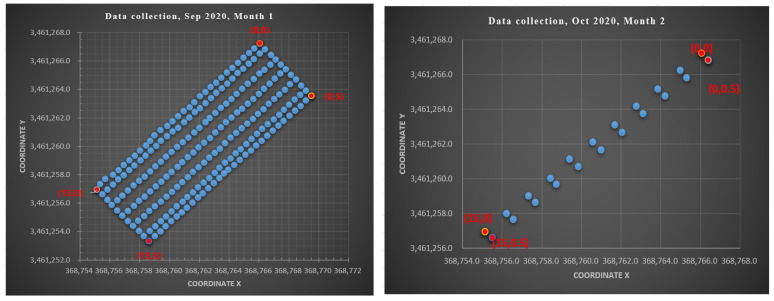
Distribution of the reference points for generating CSI-based fingerprints during September and October 2020.

**Figure 6 sensors-22-08720-f006:**
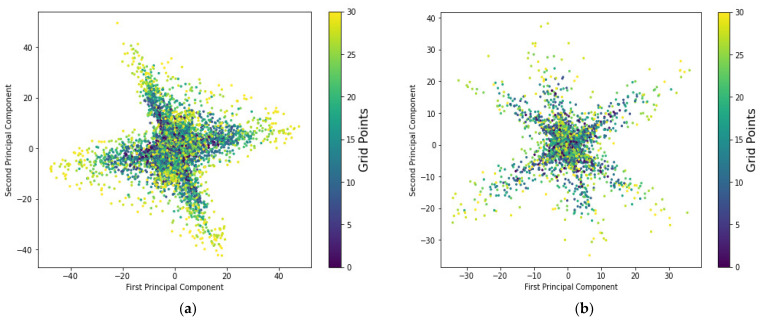
Distribution of principal components of datasets collected during month 1 with their corresponding RPs. (**a**) Training dataset Day1M1, (**b**) Testing dataset Day1M1, (**c**) Training dataset Day1M1, (**d**) Testing dataset Day1M1.

**Figure 7 sensors-22-08720-f007:**
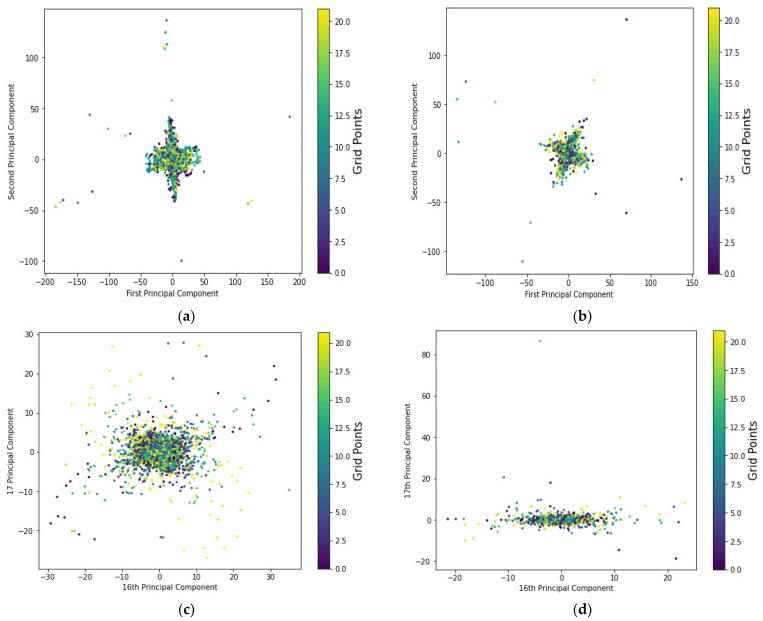
Distribution of principal components of datasets collected during month 2 with their corresponding RPs. (**a**) Training dataset Day1M2, (**b**) Testing dataset Day1M2, (**c**) Training dataset Day1M2, (**d**) Testing dataset Day1M2.

**Figure 8 sensors-22-08720-f008:**
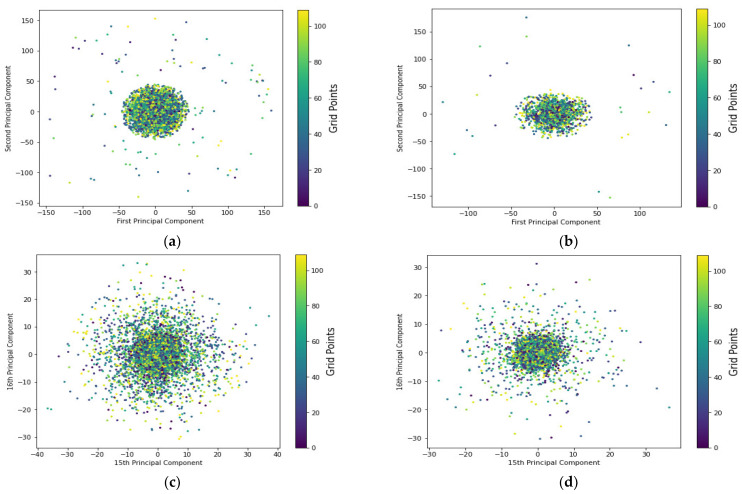
Distribution of principal components of fused data of month 2 with their corresponding RPs. (**a**) Training Fused data of Month 2, (**b**) Testing Fused data of Month 2, (**c**) Training Fused data of Month 2, (**d**) Testing Fused data of Month 2.

**Figure 9 sensors-22-08720-f009:**
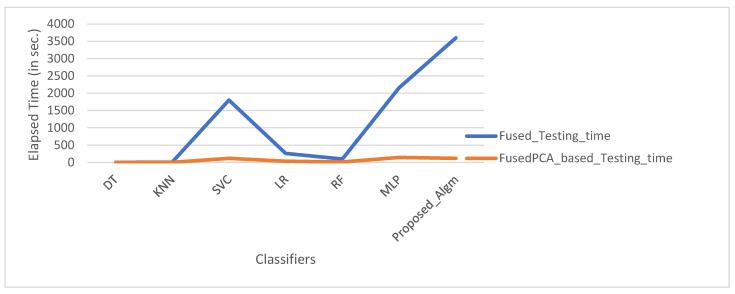
Comparative analysis of computational testing time of the fused data before and after PCA applied.

**Figure 10 sensors-22-08720-f010:**
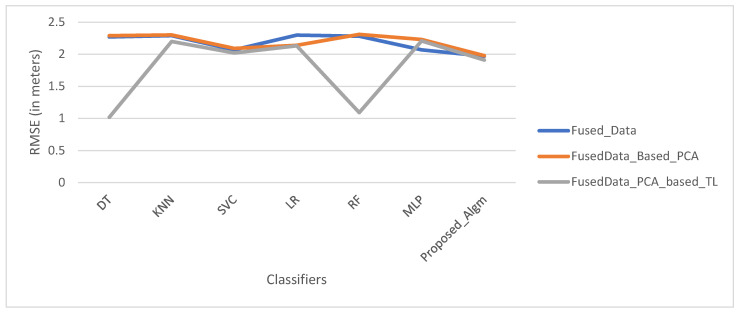
Performance analysis of classifiers applied to indoor parking lots based on fused data of CSI-fingerprints.

**Table 1 sensors-22-08720-t001:** List of Notations.

Notation	Description
Y, X, H, and ϕ	Received signal vector, transmitted signal vector, channel matrix and the AGWN (additive Gaussian white noise)
$n_{t}$$,\;n_{s},\;$$n_{T}$, $\;n_{r},$$\;\&\;n_{m}=M$	Number of target instances, number of source instances, number of transmitter antennas, number of receiver antennas, and number of subcarriers
$\hat{H}$ $,\;H^{k}(f_{m})=H_{m}$	Estimated channel frequency response (CFR) in frequency domain of all subcarriers, the channel state information for each sample of the $m^{\mathrm{th}}$ subcarrier of $k^{\mathrm{th}}$ transmitter-receiver pair.
$\left|H_{m}\right|$ $,\;∠H_{m}$	Amplitude of the $m^{\mathrm{th}}$ subcarriers, phase of the $m^{\mathrm{th}}$ subcarrier
d $,\;CSI_{eff}$	Propagation distance between the transceiver, effective channel state information measurements
c$,\;f_{c},\;$n, and φ	Radio wave phase velocity, central frequency of CSI, path loss attenuation factor, and environmental factor.
h(t) $,\;a_{i}$ $,\;\theta_{i}$ $,\;\tau_{i},\;$ N and δ(t)	Channel impulse response (CIR), amplitude values, phase, time delay of the $i^{\mathrm{th}}$ path, the total number of multipaths and Dirac delta function
$\left\{c_{rj}^{ai}(t_{d}),a\in [1,n_{a}],i\in [1,n_{c}]\right\}$	The $i^{\mathrm{th}}$ CSI amplitude value at the $r^{\mathrm{th}}$ RP of the $j^{\mathrm{th}}$ BS from $a^{\mathrm{th}}$ antenna. $t_{d}$ refers to the measurement days.
$n_{c}$ $,\;n_{a}$	Number of CSI measurements at each RP, number of antennas of a BS of a receiver
$L_{j}$ $,\;{\hat{L}}_{j}$ $,\;r_{0;j}^{2}$ $,\;\mathrm{and}\;\omega_{i^{\prime }j^{\prime }}$	The position of the $j^{\mathrm{th}}$ base station, the unknown location of the target, the distance between the $j^{\mathrm{th}}$ base station and the target, and weight transfer.
μ, ∑ $,\;\rho_{rj}^{ai}$	The overall mean values of the CSI measurements, multidimensional covariance of different sources of CSI measurements, the correlation coefficients between the two feature vectors
λ $,\;G_{Tx}$ $,\;G_{Rx}$	The wavelength of the transmitted signal, the antenna gains at the transmitter, the antenna gains at the receiver
$C_{k}^{q}\left(t_{1}\right)$ $,\;C_{k}^{q}\left(t_{2}\right)$	The fused CSI measurements collected on different measurement days of two months of September and October 2020
${C^{\prime }}_{rj}^{ai}(s_{d}){|}_{tr}$ $,\;{C^{\prime }}_{rj}^{ai}(s_{d}){|}_{ts}$ $,\;{C^{\prime }}_{rj}^{ai}(s_{tr}^{f}),\;$ ${C^{\prime }}_{rj}^{ai}(s_{ts}^{f})$	Refined Sources Fingerprint of training dataset, Refined CSI testing dataset, Fused refined CSI training fingerprints, Fused refined CSI testing fingerprints

**Table 2 sensors-22-08720-t002:** System description.

Spectrum	5.8 GHz
Bandwidth	100 MHz
Subcarrier bandwidth	120 kHz
Label position frequency	1 Hz

**Table 3 sensors-22-08720-t003:** Distributions of channel state information measurements per Label.

September 2020	Labels (#RPs = 225)	0	1	127	--	159	187	224
		**0**	**1**	**2**	**--**	**28**	**29**	**30**
**#of CSI values per Label**	Dataset 1 (#Labels = 31)	793	811	742		836	806	1183
		**31**	**32**	**33**	**--**	**34**	**35**	**36**
	Dataset 2 (#Labels = 6)	899	673	820		831	842	1017
		**37**	**38**	**39**	**--**	**59**	**60**	**61**
	Dataset 3 (#Labels = 25)	931	805	803		810	1060	1254
		**62**	**63**	**64**	**--**	**96**	**97**	**98**
	Dataset 4 (#Labels = 37)	798	841	863		930	795	806
		**99**	**100**	**101**	**--**	**132**	**133**	**134**
	Dataset 5 (#Labels = 36)	822	785	797		791	804	841
		**135**	**136**	**137**	**--**	**159**	**160**	**161**
	Dataset 6 (#Labels = 27)	870	861	994		822	799	833
		**162**	**163**	**164**	**--**	**190**	**191**	**192**
	Dataset 7 (#Labels = 31)	1009	804	801		857	792	832
		**193**	**194**	**195**	**--**	**222**	**223**	**224**
	Dataset 8 (#Labels =32)	814	804	819		799	857	832
**October 2020**	**Labels** (**#RPs = 110)**	**0**	**1**	**127**	**--**	**99**	**107**	**109**
		**0**	**1**	**2**	**--**	**19**	**20**	**21**
**#of CSI values per Label**	Dataset 1 (#Labels = 22)	928	805	875		847	1128	903
		**22**	**23**	**24**	**--**	**41**	**42**	**43**
	Dataset 2 (#Labels =22)	824	823	816		803	819	865
		**44**	**45**	**46**	**--**	**63**	**64**	**65**
	Dataset 3 (#Labels =22)	815	798	808		848	829	814
		**66**	**67**	**68**	**--**	**85**	**86**	**87**
	Dataset 4 (#Labels =22)	834	828	809		828	821	912
		**88**	**89**	**90**	**--**	**107**	**108**	**109**
	Dataset 5 (#Labels =22)	1025	891	843		810	941	1008

**Table 4 sensors-22-08720-t004:** Effect of feature spaces to the variance account of the predictive model for indoor parking lots using **Fused data: M1(#RPs = 225)**.

#PCA		Variance Explained Ratio
Training Data	Testing Data	
2	2	23.01%
16	16	95%
14	14	90%
12	12	85%
11	11	80%

**Table 5 sensors-22-08720-t005:** Effect of feature spaces to the variance account of the Predictive model for indoor parking lots using **Fused data M2 (#RPs = 110)**.

#PCA		Variance Explained Ratio
Training Data	Testing Data	
2	2	19.40%
16	16	95%
14	14	90%
12	12	85%
11	11	80%

**Table 6 sensors-22-08720-t006:** Models vehicle’s indoor parking lots of CSI-based fingerprints of various datasets with different temporal variations.

Metric: RMSE (Meter)								
Data Collection Period: Month 1, September 2020								
Classifiers	Day1 (#RPs = 31)	Day2 (#RPs = 6)	Day3 (#RPs = 25)	Day4 (#RPs = 37)	Day5 (#RPs = 36)	Day6 (#RPs = 27)	Day7 (#RPs = 31)	Day8 (#RPs = 32)
Decision tree	1.52	0.67	1.32	1.80	1.66	1.59	1.83	1.93
K-neighbour (KNN)	1.64	0.80	1.35	1.87	1.87	1.78	1.99	2.04
Support vector machine (SVC)	1.94	1.03	2.37	2.09	1.94	2.66	2.03	2.29
Logistic regression (LR)	2.21	1.00	2.04	2.29	2.21	2.25	2.25	2.34
Random forest	1.43	0.60	1.18	1.60	1.51	1.49	1.75	1.81
Neural network (MLP)	1.73	0.59	1.75	2.14	2.07	1.65	2.18	2.40
The proposed algorithm	1.71	0.81	1.53	1.95	1.85	1.75	1.88	1.96

Dayi (#RPs) …> represents the list of days when measurements were being taken with the corresponding number of reference points.

**Table 7 sensors-22-08720-t007:** Models vehicle’s indoor parking lots of CSI-based fingerprints of various datasets with different temporal variations.

After TL: Metric: RMSE (Meter)								
Data Collection Period: Month 1, September 2020								
Classifiers	Day1 (#RPs = 31)	Day2 (#RPs = 6)	Day3 (#RPs = 25)	Day4 (#RPs = 37)	Day5 (#RPs = 36)	Day6 (#RPs = 27)	Day7 (#RPs = 31)	Day8 (#RPs = 32)
Decision tree	0.68	0.31	0.59	0.80	0.74	0.71	0.8178	0.86
K -neighbour (KNN)	1.50	0.73	1.22	1.71	1.68	1.61	1.82	1.88
Support vector machine (SVC)	1.88	0.99	2.30	2.02	1.87	2.58	1.97	2.23
Logistic regression (LR)	2.21	0.88	1.94	2.21	2.13	2.15	2.17	2.27
Random forest	0.67	0.29	0.55	0.76	0.73	0.70	0.82	0.86
Neural network (MLP)	1.62	0.36	1.69	2.11	2.06	1.57	2.16	2.38
The proposed algorithm	1.64	0.75	1.48	1.92	1.81	1.69	1.81	1.91

Dayi (#RPs) …> represents the list of days when measurements were being taken with the corresponding number of reference points.

**Table 8 sensors-22-08720-t008:** Models vehicle’s indoor parking lots of CSI-based fingerprints of various datasets with different temporal variations.

Metric: RMSE (Meter)						
Data Collection Period: Month 2, October 2020						
Classifiers	Day1 (#RPs = 22)	Day2 (#RPs = 22)	Day3 (#RPs = 22)	Day4 (#RPs = 22)	Day5 (RP#22)	Fuseddata (#RPs = 110)
Decision tree	1.30	1.71	1.67	1.37	1.58	2.27
K -neighbour (KNN)	1.39	1.91	1.74	1.47	1.76	2.29
Support vector machine (SVC)	1.65	2.32	2.20	1.84	1.95	2.07
Logistic regression (LR)	1.89	2.39	2.19	1.88	2.25	2.30
Random forest	1.08	1.51	1.42	1.21	1.42	2.28
Neural network (MLP)	1.19	1.76	1.60	1.46	1.96	2.07
The proposed algorithm	1.52	2.02	1.76	1.53	1.80	1.97

Dayi (#RPs) …> represents the list of days when measurements were being taken with the corresponding Number of Reference points.

**Table 9 sensors-22-08720-t009:** Models vehicle’s indoor parking lots of CSI-based fingerprints of various datasets with different temporal variations.

After TL: Metric: RMSE (Meter)					
Data Collection Period: Month 2, October 2020					
Classifiers	Day1 (#RPs = 22)	Day2 (#RPs = 22)	Day3 (#RPs = 22)	Day4 (#RPs = 22)	Day5 (#RPs = 22)
Decision tree	0.57	0.75	0.75	0.61	0.70
K -neighbour (KNN)	1.26	1.72	1.60	1.33	1.62
Support vector machine (SVC)	1.60	2.26	2.09	1.77	1.87
Logistic regression (LR)	1.79	2.26	2.11	1.79	2.14
Random forest	0.52	0.75	0.67	0.56	0.68
Neural network (MLP)	1.08	1.56	1.49	1.35	1.91
The proposed algorithm	1.47	1.92	1.70	1.47	1.72

Dayi (#RPs) …> represents the list of days when measurements were being taken with the corresponding number of reference points.

**Table 10 sensors-22-08720-t010:** Models Vehicle’s indoor parking lots of CSI-based fingerprints of **fused data** based PCA method.

After TL: Metric: RMSE (Meter)		
Data Collection Period: Month 2, October 2020		
Classifiers	95% Explained Variance Ratio (EVR)	
	Day5 (#RPs = 22)	Fused Data (#RPs = 110)
Decision tree	14.69	2.29
K -Neighbour (KNN)	15.09	2.30
Support vector machine (SVC)	12.56	2.09
Logistic Regression (LR)	12.51	2.14
Random forest	15.73	2.31
Neural Network (MLP)	14.30	2.23
The proposed algorithm	1.78	1.91

TL …> represents transfer learning.

**Table 11 sensors-22-08720-t011:** Parameters specification used for each algorithm.

SNo.	Parameters	Descriptions
Decision tree	Default	DecisionTreeClassifier
K -Neighbour (KNN)	n_neighbors = 20	
Support vector machine (SVC)	kernel = ‘poly’, C = 0.6	C-regularization parameter
Logistic Regression (LR)	Default	
Random forest (RF)	n_estimators = 6	Random Forest Tree
Neural Network (MLP)	MLPClassifier, random_state = 10, max_iter = 500	Multi-layer Perceptron classifier

## Data Availability

The dataset used for this study are available upon request to the corresponding author.

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
