# Peer review of "Data Fusion Methods for Indoor Positioning Systems Based on Channel State Information Fingerprinting"

_sensors, 2022, doi:10.3390/s22228720_

Round 1

Reviewer 1 Report

The paper is interesting and the content is adequately innovative. An indoor positioning method is based, relying on channel state information acquired from WiFi transmissions to advantageously replace RSS measurements in a fingerprinting based approach.
The conclusions are interesting and the presented work is credible, however Authors should state more clearly that the accuracy is improved with respect to RSS based systems, and computational cost is improved with respect to other positioning algorithms. In its current form the paper technical quality is significantly limited by the presentation of the theoretical background. The paper should be thoroughly revised by the Authors to remove this weakness and then checked again. To this aim, a list of specific remarks is listed in the following as a guidance

Line 131: typo, "anlaysis" shuld read "analysis", also on line 279

Line 140: the "scene analysis" method is not seemingly explicitly described in [44] and should be described more deeply in this paper.

Lines 208-210: X,Y, and H should be defined explicitly upon their first occurrence. What is the dimension of H? Are X and Y phasors vectors obtained for a set of given frequencies?

Line 216: defining the size of H as n_t*n__r*n_m is a bit misleading, since it suggests a 3D matrix, for which a matrix product is replaced by a tensor reduction.

Line 219, Equation (3): please check whether f_s should read f_m.

Line 223, Equation (4): it is not clear whether Hm is H^k_m defined in line 220 or if it is something else. If the two symbols describe the same quantity, a single symbol may be used.

Line 229: the acronym FILA (Fine Grain Indoor Localization) should be defined upon its first occurrence.

Line 233, it is not clear how the norm ||A||m relates to H_m.

Line 236, "radio velocity" should read "speed of light" or "radio wave phase velocity".

Line 237, the meaning of the environmental factor sigma should be explained.

Line 242, Equation (9) is seemingly incorrect as it depicts the Tapped Delay Line model, where the Dirac delta function is not part of the exponential.

Lines 247-250, Equations (10)(11): the phase term phi has disappeared in (10), reappearing in (11)

Lines 290-293, the sentence is very unclear, possibly missing one or two verbs.

Fig. 1: if the data preprocessing algorithm is fed by readings, please consider revising the figure according to the data flow.

Line 322: the symbol n_c is seemingly undefined.

Line 323: n is redefined, since it was used in line 237 and 264 to designate the path loss.

Line 334: the summation upper end "b" is undefined

Line 344: please check whether n_t redefines the same symbol used in line 216. If so, please consider using a different symbol.

Line 375, in introducing Eq. (18) please consider motivating the use of a Gaussian model

Line 411, Equation (23) the summation index i does not appear in the summation term.

Line 412: Equation (23) seemingly suggests that the Manhattan distance is being used, so it is not clear why the Euclidean distance is being mentioned.

Author Response

Reviewer #1:

Notes on Revision

Re: Manuscript sensors-194-8894, entitled Data Fusion Methods in Indoor Positioning Systems Based on Channel State Information Fingerprinting

We would like to express our gratitude to the editors and the anonymous reviewers for their constructive suggestions and criticism. The comments are well taken, and the manuscript has been revised accordingly. Below please find our responses to the reviewers’ comments. Also, for the reviewers’ convenience, major changes are written in YELLOW in the revised manuscript.

Responses to the Comments of Reviewer 1

We thank the Reviewer for the comments, suggestions and questions that helped us to improve the quality of the manuscript.

The paper is interesting and the content is adequately innovative. An indoor positioning method is based, relying on channel state information acquired from WiFi transmissions to advantageously replace RSS measurements in a fingerprinting based approach.
The conclusions are interesting and the presented work is credible, however Authors should state more clearly that the accuracy is improved with respect to RSS based systems, and computational cost is improved with respect to other positioning algorithms. In its current form the paper technical quality is significantly limited by the presentation of the theoretical background. The paper should be thoroughly revised by the Authors to remove this weakness and then checked again. To this aim, a list of specific remarks is listed in the following as a guidance

Comment #1: Line 131: typo, "anlaysis" shuld read "analysis", also on line 279

Response #1: Thank you for your suggestions. In the revised version, we have corrected it.

Comment #2: Line 140: the "scene analysis" method is not seemingly explicitly described in [44] and should be described more deeply in this paper.

Response #2: Thank you for your suggestions. In the revised version, we have added the following as:

On the other side, most indoor localization technologies based on Wi-Fi rely on received signal strengths and can be directly implemented using the existing wireless communications infrastructure without any calibrations. Wi-Fi received signal strength (RSSI) (measured in decibel milliwatts: dBm) is used to find a relationship between transceivers or measures the accuracy of localization based on the distances between the mobile user and the available Wi-Fi access points [47] – [49] through the third method so-called the Scene analysis comprises two phases: training and testing phases. The RSS fingerprints the so-called radio map are first collected from each Wi-Fi access point at multiple locations within the defined grid points (GPs) and a predictive model is trained to learn the ‘signal-to-location’ relationship (training phase). The learned model is then applied to infer the location of the target based on the new measurement obtained (online phase) [47] – [49].

Comment #3: Lines 208-210: X,Y, and H should be defined explicitly upon their first occurrence. What is the dimension of H? Are X and Y phasors vectors obtained for a set of given frequencies?

Response #3: Thank you for your constructive suggestion. In this revised version, we have added the following Table to better understand:

Table 1: List of Notations

Notation

Description

, , , and

Received signal vector, transmitted signal vector, channel matrix and the AGWN (additive Gaussian white noise)

,   ,  and

Number of target instances, number of source instances, number of transmitters, number of receiver antennas and the number of subcarriers

,

Estimated channel frequency response (CFR) in frequency domain of all sub carriers, the channel state information for each sample of the  subcarrier of transmitter-receiver pair.

,

Amplitude of the  subcarriers, phase of the  subcarrier

  ,

Propagation distance between the transceiver, effective channel state information measurements,

, , , and

Radio wave phase velocity, central frequency of CSI, path loss attenuation factor, and the environmental factor.

, , , , and

Channel Impulse Response (CIR), amplitude values, phase, and time delay of the  path, the total number of multipaths and Dirac delta function

The  CSI amplitude value at the  RP of the  BS from antenna.  refers the measurement days.  and  are the number of CSIs collected from an antenna of a BS and number of antennas of a BS respectively.

, , , and

The position of the base station, the unknown location of the target, the distance between the base station and the target, and weight transfer.

,

The overall mean values of the CSI measurements, multidimensional covariance of different sources of CSI measurements, the correlation coefficients between the two feature vectors

, ,

The wavelength of the transmitted signal, the antenna gains at the transmitter, the antenna gains at the receiver

Comment #4: Line 216: defining the size of H as n_t*n__r*n_m is a bit misleading, since it suggests a 3D matrix, for which a matrix product is replaced by a tensor reduction.

Response #4: Thank you for your suggestion and we share your concern but the physical layer CSI over multiple sub-carriers with the dimension basically reveals the number of transmitter ( ), receiver antennas ( ) and the number of subcarriers ( ) for each antenna pair respectively.

Comment #5: Line 219, Equation (3): please check whether f_s should read f_m.

Response #5: Thank you for your comment and we have corrected it accordingly. It is f_m.

Comment #6: Line 223, Equation (4): it is not clear whether Hm is H^k_m defined in line 220 or if it is something else. If the two symbols describe the same quantity, a single symbol may be used.

Response #6: Thank for your constructive suggestion and in the revised version we have corrected it as:  and we used  for simplicity after we defined .

Comment #7: Line 229: the acronym FILA (Fine Grain Indoor Localization) should be defined upon its first occurrence.

Response #7: Thank you for comment and in the revised version we have corrected it accordingly.

Comment #8: Line 233, it is not clear how the norm ||A||m relates to H_m.

Response #8: Thank you so much and we have corrected it as:

The norm symbol is used by mistake to represent the magnitude symbol. We thank you again.

Comment #9: Line 236, "radio velocity" should read "speed of light" or "radio wave phase velocity".

Response #9: Thank you for the constructive suggestion and we have corrected it. We have used radio wave phase velocity.

Comment #10: Line 237, the meaning of the environmental factor sigma should be explained.

Response #10: Thank you for your suggestion. We have added the following statement:

The environment setting was being conducted in a specified area in controlled fashion, and assumed a constant environmental factor ( ) to estimate the distance between the transceiver. Moreover,  is the path loss attenuation factor. The idea behind the environmental factor describes that the targets are exposed or shared same experiences within the experimental setting of the defined indoor parking region as a baseline though practically difficult to ensure it.

Comment #11: Line 242, Equation (9) is seemingly incorrect as it depicts the Tapped Delay Line model, where the Dirac delta function is not part of the exponential.

Response #11: Thank you for your valuable suggestion. We checked and corrected it as follows:

                                           (9)

Comment #12: Lines 247-250, Equations (10)(11): the phase term phi has disappeared in (10), reappearing in (11)

Response #12: Thank you for your valuable suggestion. We checked and corrected it as follows:

The CIR is characterized as Channel Frequency Response (CFR) in the frequency domain and it has been reported [69] in the commercial off-the-shelf WiFi devices that sampled versions of CFRs are revealed to upper layers in the format of CSI. Thus,

                                           (10)

Where  is the frequency of  subcarrier and  is CSI at  subcarrier and each CSI depicts the amplitude and phase of a subcarrier as:

                                             (11)

Comment #13: Lines 290-293, the sentence is very unclear, possibly missing one or two verbs.

Response #13: Thank you for your constructive suggestions and we have rewritten as follows:

By analyzing the limitations of signal strength values (RSSI) fingerprint locations, geometric locations, and inertial navigation locations, an indoor data fusion method based on an adaptive unscented Kalman filter (UKF) was proposed [75]. The algorithm uses a six-position error calibration method and Kalman filter to compensate for the MEMS-SINS data and establishes the correlation between location data and RSSI/geomagnetic data based on the feature sorting vector fingerprint matching method which leads to improved data stability and indoor location accuracy [75].

Comment #14: Fig. 1: if the data preprocessing algorithm is fed by readings, please consider revising the figure according to the data flow.

Response #14: Thank you for the comment. But, Fig. 1 already depicts the schematic representation of the proposed framework of CSI-based Fingerprint Data Fusion technique for Indoor Positioning System.

Comment #15: Line 322: the symbol n_c is seemingly undefined.

Response #15: Suppose  denote the  CSI amplitude value at the  RP of the  BS from antenna. The   refers the time when the data were collected specifically measured in number of days.  and  are the number of CSIs collected from an antenna of a BS and number of antennas of a BS respectively.

Comment #16: Line 323: n is redefined, since it was used in line 237 and 264 to designate the path loss.

Response #16: Thank you for your suggestion. We have corrected it.

Comment #17: Line 334: the summation upper end "b" is undefined

Response #17: Thank you for your suggestion and we have corrected as there were detectable base stations  in total.

Comment #18: Line 344: please check whether n_t redefines the same symbol used in line 216. If so, please consider using a different symbol.

Response #18: n_t  and n_s denotes number of target and source instances respectively. They are different.

Comment #19: Line 375, in introducing Eq. (18) please consider motivating the use of a Gaussian model.

Response #19: Each vector of the CSI amplitude measurements is assumed to be independent and identically distributed such that the joint probability density function with  dimension of random vectors of the CSI amplitude values observed certainly on different temporal variations. This is why we have assumed a multidimensional covariance matrix to consider the temporal effect of separate measurement days.

Comment #20: Line 411, Equation (23) the summation index i does not appear in the summation term.

Response #20: Thank you for your comment and in the revised version, we have corrected it.

Comment #21: Line 412: Equation (23) seemingly suggests that the Manhattan distance is being used, so it is not clear why the Euclidean distance is being mentioned.

Response #21: We specified that and denotes Manhattan and Euclidean distances respectively. We have used for our case. The new feature vectors were used to minimize the variations of the amplitude values of CSI based fingerprints and indicated in equation (24).

Reviewer 2 Report

Using CSI for indoor positioning is now well known with a large amount of published results. The main novelty in this paper is the transfer learning but is not explained in a simple and understandable way nor with available and published references.

The quality of the paper must be improved for a better understanding. Too many notations are confusing and a lot of equations need more details and steps to be verified. 

Ref 37 is needed for detailed explanations, without it is not possible to understand the TL algorithm considered.

Title : “Data Fusion Methods for Indoor Positioning Systems based Channel State Information Fingerprinting” --> Data Fusion Methods for Channel State Information Fingerprinting based Indoor Positioning Systems ?

Abstract : please make your sentences shorter, this is a general recommendation also for the other sections of the paper.

Line 82 : standard deviation unit for 16.8, 15.9, 14.5, 17.9 . .?

Line 169 : RSS is also highly dependent on the used Wi-Fi chipset and how it estimates and reports the RSS value.

Line 210 : use same typo for X,Y,H,..

Line 223 : eq (4), what is k ? index k ? why sin( ) ?   wrong formula.

Line 231 : eq(6), sum over i=1 ..K, I don’t understand what represents index ‘i’ in your sum. Precise the definition of A.

Line 234 : the expression you give for d is an estimator based on a log distance path loss model, it is not the true value of the Tx-Rx distance

Line 242 : typo, the dirac is not in the exponent

Line 247 and 256:  error, the phase theta_i of each path is missing

Line 269 : eq 16, how do you justify the same environment factor 'n' for all the paths ? why  would it be the same for LOS and NLOS paths?

Line 292: grammar ? your sentence can’t start with “and”

Line 293 : grammar ?, correct “ the algorithm was used”

Line 319 : “.. fingerprints database involves two main phases..” : I don’t understand, seems words are missing

Line 322 : do you consider only CSI amplitude and not the phase in your IPS algorithm ?

What is ‘nc’, is it M ? I suggest to add a table to define your notations and keep the same notation and definition all over the article.

Line 333: definitions of td and d1, … dn are unclear, to be better explained

Lines 334/344 : avoid confusing notations in the text it is very unclear if the same notation C** designates a sample or the whole set of data

Line 372 : ‘CSI amplitude measurements are assumed to be independent and identically distributed’ : it is a strong hypothesis, this needs more explanations. In particular you have a MIMO system, you need at least to have similar antennas at Tx and also at Rx side. It’s probably true for a GWSSUS channel, but from one day to another day you may have additional obstacles (vehicles) and so another CSI normal density with other mean and std values.

Line 399 : unclear, do you refer to the fact that there is a risk to have the same CSI fingerprint for 2 different RPs ?

Line 411 : eq 23, where do you consider || ||1 and || ||2  ? , what represents i’ index in the sum ?

Line 416 :  lessor ? à lessen

Line 419 : need more explanations or available references to detail how the transfer coefficients are then used to modify the data features. Are you considering manifold alignment algorithm ? This is the main novelty of the paper and it needs much more details and explanations.

Line 421 : this needs more explanations, how do you deduce location estimates ?

Line 433 / step 15 : ‘until convergence’ --> which convergence, which criteria ?

Line 447 : grammar error “estimate” à estimates + sentence too long

Line 526-528 : eq 45-46 need more explanations, or detailed in annex. I don’t understand why you consider G or ∑ which concerns CSI and not (xi,yi).

Table 5: number of layers, nodes, type full dense, CNN, .. ? Neural Network is too generic.

Line 593 section 4 : data sets are recorded at different days, but the key point is the parking occupancy and its variations among the measurement days. Could you precise this point ?  

You consider RP, but can you detail if your prediction consists into classifiers only or into regression ?

Author Response

Reviewer #2

Notes on Revision

Re: Manuscript sensors-194-8894, entitled Data Fusion Methods in Indoor Positioning Systems Based on Channel State Information Fingerprinting

We would like to express our gratitude to the editors and the anonymous reviewers for their constructive suggestions and criticism. The comments are well taken, and the manuscript has been revised accordingly. Below please find our responses to the reviewers’ comments. Also, for the reviewers’ convenience, major changes are written in YELLOW in the revised manuscript.

Responses to the Comments of Reviewer 2

We thank the Reviewer for the comments that helped us to improve the quality of the manuscript.

Comment #1: Ref 37 is needed for detailed explanations, without it is not possible to understand the TL algorithm considered.

Response #1: Thank you for your valuable suggestion. The main idea behind Ref [37] can be described as “an online heterogeneous transfer learning (OHetTLAL) algorithm was proposed for indoor positioning system -based RSS fingerprinting to improve the positioning performance in the target domain by fusing both source and target domains knowledge. To that end, a new feature spaces was derived (on which the model is trained) based on the co-occurrence of RSS measurements from mobile devices in the two domains in order to capitalize on knowledge that could improve target location prediction. Thus, the higher the value of co-occurrence between the two domains, the more detected Wi-Fi APs are shared by these two domains and the more likely these source domains are related to the target domain, which positively affects the prediction of the target.

Comment #2: Title: “Data Fusion Methods for Indoor Positioning Systems based Channel State Information Fingerprinting” --> Data Fusion Methods for Channel State Information Fingerprinting based Indoor Positioning Systems?

Response #2: Thank you for your comment. We have corrected as: Data Fusion Methods for Indoor Positioning Systems Based on Channel State Information Fingerprinting”

Comment #3: Abstractplease make your sentences shorter, this is a general recommendation also for the other sections of the paper.

Response #3: In the original version, we rewrite the abstract as:

“Despite decades of advancements in location services, radio signals propagating indoors face non-line-of-sight (NLOS) propagation effects, multipath effects, and a dynamic environment that presents more challenges than outdoor conditions. Modern Wi-Fi networks that are able to use both MIMO-OFDM techniques have emerged as Channel State Information (CSI) as an enhanced wireless channel metric with significant data throughput, replacing Wi-Fi received signal strength (RSS)-based fingerprinting. However, it is still not robust and unstable due to its multipath effects. In this paper, a positive knowledge transfer-based heterogeneous data fusion method is developed to represent the different scenarios in which CSI-based fingerprinting measurements are generated in complex indoor environments targeting indoor parking lots while reducing training calibration overhead. The proposed algorithm was extensively tested against real-world scenarios of indoor parking, and the results showed that it proved to be an efficient algorithm with consistent positioning accuracy across all possible variations. Not only does the proposed algorithm improve indoor parking location accuracy, but it also is a computationally robust and efficient estimator of dynamic indoor environments. Cramer Rao Lower Bound (CRLB) analysis was also used to estimate the lower bound of the variance of the parking lot location error under various temporal variation scenarios. Based on analytical derivations, the lower bound of the variance of the location estimator depends on i) the angle of the base stations, ii) the number of base stations, iii) the distance between the target and the base station,  iv) the correlation of the measurements, and v) the signal propagation parameters  and .”

Comment #4: Line 82: standard deviation unit for 16.8, 15.9, 14.5, 17. 9...?

Response #4: We have corrected it and the standard deviation unit is dBm (measured in decibel milliwatts: dBm).

Comment #5: RSS is also highly dependent on the used Wi-Fi chipset and how it estimates and reports the RSS value.

Response #5: Thank you for your valuable comment. We have added the following text as: Moreover, RSS is also highly dependent on the used Wi-Fi chipset and how it estimates and reports the RSS value.

Comment #6: Line 210: use same typo for X, Y, H,…,…

Response #6: In the original version, we have added the following new Table.

Table 1: List of Notations

Notation

Description

, , , and

Received signal vector, transmitted signal vector, channel matrix and the AGWN (additive Gaussian white noise)

,   ,  and

Number of target instances, number of source instances, number of transmitters, number of receiver antennas and the number of subcarriers

,

Estimated channel frequency response (CFR) in frequency domain of all sub carriers, the channel state information for each sample of the  subcarrier of transmitter-receiver pair.

,

Amplitude of the  subcarriers, phase of the  subcarrier

  ,

Propagation distance between the transceiver, effective channel state information measurements,

, , , and

Radio wave velocity, central frequency of CSI, path loss attenuation factor, and the environmental factor.

, , , , and

Channel Impulse Response (CIR), amplitude values, phase, and time delay of the  path signal propagation, the total number of multipaths and Dirac delta function

The  CSI amplitude value at the  RP of the  BS from antenna.  refers the measurement days.  and  are the number of CSIs collected from an antenna of a BS and number of antennas of a BS respectively.

, , , and

The position of the base station, the unknown location of the target, the distance between the base station and the target, and weight transfer.

Comment #7: Line 223 : eq (4), what is k ? index k ? why sin( ) ?   wrong formula. Thank you, we have corrected as:

Response #7: Thank you for your valuable suggestion. We have corrected as:

                                              (4)

Where,  and represents the amplitude and phase of  subcarriers respectively.

Comment #8: Line 231 : eq(6), sum over i=1 ..K, I don’t understand what represents index ‘i’ in your sum. Precise the definition of A. Thank you so much for correcting us.

Response #8: In the original version, we have corrected as:

                          (6)

Where  is the effective CSI for distance estimation, and  are the number of subcarriers and the calculated center frequency, and  is the amplitude of the filtered CSI on the  subcarrier. In addition,  which the index i represents the transmitter-receiver pair.

Comment #9: Line 234 : the expression you give for d is an estimator based on a log distance path loss model, it is not the true value of the Tx-Rx distance.

Response #9: Thank you for your valuable suggestion. Yes, it is an estimated distance based on a log distance path loss model and corrected it accordingly.

Comment #10: Line 242: typo, the dirac is not in the exponent

Response #10: Thank you for your valuable suggestion. We checked and corrected it as follows:

                                           (9)

Comment #11: Line 247 and 256:  error, the phase theta_i of each path is missing.

Response #11: Thank you for your comment. We have corrected it as:

                                             (11)

where  is the amplitude and is the phase of each subcarrier and the received signal strength (in dBm) at each subcarrier is proportional to the amplitude of CSI.

Comment #12: Line 269 : eq 16, how do you justify the same environment factor 'n' for all the paths ? why  would it be the same for LOS and NLOS paths?

Response #12: Thank you for your suggestion. Well it is subjective. There are various parameters in the equation that really could bring variations with in a given environment. So, as long as the environment setting is being conducted in a specified area in controlled fashion, a constant environmental factor ( ) is assumed. Moreover,  is the path loss attenuation factor. The environment factor is ( ) while  is the path loss attenuation factor. We think exposing the testing units to same baseline for comparative purpose.

Comment #13: Line 292: grammar ? your sentence can’t start with “and”. Thanks, we have corrected.

Response #13: Thank you for your valuable suggestion. We have corrected it.

Comment #14: Line 293 : grammar ?, correct “ the algorithm was used”. Thanks, we have corrected it.

Response #14: Thank you for your valuable suggestion. We have corrected it.

Comment #15: Line 319 : “.. fingerprints database involves two main phases..” : I don’t understand, seems words are missing

Response #15: Thank you for your constructive suggestion. In this revised version, we have rewrite it as:

In this paper, as depicted in Fig. 2 below we adopted the second method so called the Scene analysis to construct a database of fingerprints which comprises two main phases: training and testing phases.

Comment #16: Line 322 : do you consider only CSI amplitude and not the phase in your IPS algorithm ?

Response #16: Thank you for your question. Yes.

Comment #17: What is ‘nc’, is it M ? I suggest to add a table to define your notations and keep the same notation and definition all over the article.

Response #17: Thank you for your constructive suggestion. In this revised version, we have added the following Table to better understand:

Table 1: List of Notations

Notation

Description

, , , and

Received signal vector, transmitted signal vector, channel matrix and the AGWN (additive Gaussian white noise)

,  and

Number of transmitters, number of receiver antennas and the number of subcarriers

,

Estimated channel frequency response (CFR) in frequency domain of all sub carriers, the channel state information for each sample of the  subcarrier of transmitter-receiver pair.

,

Amplitude of the  subcarriers, phase of the  subcarrier

  ,

Propagation distance between the transceiver, effective channel state information measurements,

, , , and

Radio wave velocity, central frequency of CSI, path loss attenuation factor, and the environmental factor.

, , , , and

Channel Impulse Response (CIR), amplitude values, phase, and time delay of the  path, the total number of multipaths and Dirac delta function

The  CSI amplitude value at the  RP of the  BS from antenna.  refers the measurement days.  and  are the number of CSIs collected from an antenna of a BS and number of antennas of a BS respectively.

, , , and

The position of the base station, the unknown location of the target, the distance between the base station and the target, and weight transfer.

Comment #18: Line 333: definitions of td and d1, … dn are unclear, to be better explained.

Response #18: Thank you for your constructive suggestion. d1, … dn represents the number of days that CSI measurements are taken.

Comment #19: Lines 334/344 : avoid confusing notations in the text it is very unclear if the same notation C** designates a sample or the whole set of data.

Response #19: Thank you for your constructive suggestion. In this revised version, we have added a new Table to better understand.

Comment #20: Line 372 : ‘CSI amplitude measurements are assumed to be independent and identically distributed’ : it is a strong hypothesis, this needs more explanations. In particular you have a MIMO system, you need at least to have similar antennas at Tx and also at Rx side. It’s probably true for a GWSSUS channel, but from one day to another day you may have additional obstacles (vehicles) and so another CSI normal density with other mean and std values.

Response #20: Thank you for your constructive suggestion. We have similar idea such that each vector of the CSI amplitude measurements is assumed to be independent and identically distributed such that the joint probability density function with  dimension of random vectors of the CSI amplitude values observed certainly on different temporal variations. This is why we have assumed a multidimensional covariance matrix to consider the temporal effect of separate measurement days.

Comment #21: Line 399 : uncleardo you refer to the fact that there is a risk to have the same CSI fingerprint for 2 different RPs ? Yes, of course.

Response #21: Thank you for your question. Yes, of course.

Comment #22: Line 411 : eq 23, where do you consider || ||1 and || ||2  ? , what represents i’ index in the sum ?

Response #22: Thank you for your constructive suggestion. i- the csi measurements collected from the reference points. We used || ||2 in our case. There are various scenarios to consider among the two such as outlier distribution of your dataset, measurements distribution, and sparsity of your data. We suggest that the accuracy value should not be the only criterion used to select a loss function in general. In some cases, how significant is the gain given that other parameters may also be a factor? In most practical scenarios, a single metric does not provide a stable and significant accuracy score instead other methods including effective data preprocessing, handling missing values, handling imbalance data problem, irrelevant features, need to be seriously considered and addressed in our analysis for robust estimation.

Comment #23: Line 416 :  lessor ? à lessen

Response #23: Thank you for your comment. We corrected as:The objective function's equality constraint would assign higher weights to the most related source instances and lower weights to the least related source instances.”

Comment #24: Line 419 : need more explanations or available references to detail how the transfer coefficients are then used to modify the data features. Are you considering manifold alignment algorithm ? This is the main novelty of the paper and it needs much more details and explanations.

Response #24: Thank you for your constructive suggestion. Conceptually, we are trying to extract a projection to space of lower dimensionality preserving the values of distance between all the data pairs. However, unlike the manifold which most used for nonlinear methods, we adopt PCA instead which is a linear approach. Moreover, the objective function's equality constraint would assign higher weights to the most related source instances and lower weights to the least related source instances. The new feature vectors were used to minimize the variations of the amplitude values of CSI based fingerprints and the transfer coefficients can be estimated using the Lagrangian multiplier method which is detained in equation (24)- (28).

Comment #25: Line 416 : Line 421 : this needs more explanations, how do you deduce location estimates ?

Response #25: Thank you so much for helping us to clarify the matter. In the revised version, we have added the following statement to describe the detailed procedure being used to meet the goal of prediction:

“CSI fingerprints are collected from each Wi-Fi access point at multiple locations within the defined reference points (RPs) during the training phase, and a predictive model is trained to characterize the "signal-to-location" learning relationship (offline phase). During the testing phase, the learned model is then used to predict the location of the mobile user based on the new CSI measurement received.”

Comment #26: Line 433 / step 15: ‘until convergence’ --> which convergence, which criteria?

Response #26: Thank you so much for helping us to clarify the matter. In the revised version, we have rewritten the steps as follows:

Algorithm 2.  The proposed heterogenous knowledge transfer-based CSI-fingerprint Indoor Positioning system

1. Input: Refined Sources Fingerprint

2. Input: Refined CSI testing dataset 

3. Output: Fused refined CSI Fingerprints  position estimates, transfer coefficients

4. for  do

5.    for  do

6.         for  do

7.                    for  do

8. repeat

9.      Step 1:

10.       Fuse the CSI amplitude fingerprints from all the sources with temporal variations as in Algorithm 1.     11.      Step 2:

12               Obtain projection matrix:

13.      Step 3:

14.               Compute  by using equations (24-28)

15.                           end for

16.           end for

17.       end for

18. end for

19. until convergence

20. Train a classifier from  with considering weights of source domains

21. Estimate Target’s location   on   by applying the trained classifier

22. return,,

Comment #27: Line 447 : grammar error “estimate” à estimates + sentence too long. Corrected.

Response #27: Thank you for your comment. We have corrected it and shortened the sentence.

Comment #28: Line 526-528 : eq 45-46 need more explanations, or detailed in annex. I don’t understand why you consider G or ∑ which concerns CSI and not (xi,yi).

Response #28: Thank you for your comment. G is a fixed term and  represents the covariance coefficient matrix of the signal variations of CSI measurements collected on different days which are given as below:

 and             (44)

Comment #29: Table 5: number of layers, nodes, type full dense, CNN, .. ? Neural Network is too generic.

Response #29: Thank you for your comment. In the revised version, the following Table 10 is added to better understand parameters specifications being used by the algorithms in the paper.

Table 10. Parameters specification used for each algorithm

SNo.

Parameters

Descriptions

Decision tree

Default

DecisionTreeClassifier

K -Neighbour (KNN)

n_neighbors=20

Support vector machine (SVC)

kernel='poly', C = 0.6

C- regularization parameter

Logistic Regression (LR)

Default

Random forest (RF)

n_estimators = 6

Random Forest Tree

Neural Network (MLP)

MLPClassifier, random_state=10, max_iter=500

Multi-layer Perceptron classifier

Comment #30: Line 593 section 4 : data sets are recorded at different days, but the key point is the parking occupancy and its variations among the measurement days. Could you precise this point?

Response #30: Thank you for your question. Of course, yes, we share the idea that parking occupancy and its variations among measurement days is our focus too. To this end, we presented a number of real-world experiments that were carried out on various occasions or on a separate day to measure the temporal signal fluctuations-based channel state information fingerprinting. Moreover, your concern is explicitly described in Table 2. Thus, we rewrite as follows:

“This section presents a number of real-world experiments that were carried out on various occasions or on a daily basis to measure the temporal signal fluctuations using the fingerprints of the channel state information (interested in variations among the measurement days), and we assess our proposed algorithms as they were applied to these CSI real measurements of various in distributions. First, experimental conditions, datasets, and an analysis of the algorithms' overall performance were presented.”

Comment #31: You consider RP, but can you detail if your prediction consists into classifiers only or into regression?

Response #31: Thank you for your question. We considered classification.

Reviewer 3 Report

The topic of this paper is interesting and up to date, the manuscript has a practical character. Overall, it is informative and written in proper English. This is a long manuscript (30+ pages), the number of cited references is very large. However, at first glance, there seems to be an issue with the formatting and editorial side. Additionally, symbols and mathematical equations are not properly and uniformly formatted.

The experimental part is scientifically sound, the number of utilized equipment (e.g., 225 and 110 RPs) is very nice.

Suggestions and comments:

Check how to properly prepare the Authors and Affiliation section and make necessary corrections.

The title of the paper should have [based on] in its name.

Numerous mathematical symbols and equations seem oddly stretched, they are not properly formatted.

Additionally, check the spacing between subsequent lines, utilized fonts, etc.

Figures and block diagrams could be inserted in higher resolutions. Currently, many of them do not have sharp edges, they look blurred. They also lack proportion.

They way that Figure 4 was modified (labelling the axes) shows how not to do it. Those need to be reedited once again.

Other latter figures do look different. All plots/charts should be uniformly formatted. Check the style, fonts, their size, etc.

After going through the whole paper, I am not sure about the research campaign. Authors should highlight how many devices did they utilize. What kind of devices were available? What was their technical specification? Why did you select them? Why did you perform your experiments in such an environment, etc.

Furthermore, the References section is not properly and uniformly formatted.

The Conclusions part is far too short. Do provide additional feedback on your findings for the potential reader. Do mention about open issues and future study directions that might inspire other researchers. Comments are necessary.

To sum up, this is a good paper. However, Authors did not pay much attention when preparing their manuscript. Its current form lacks adequate and precise formatting.

Author Response

Reviewer #3:

Notes on Revision

Re: Manuscript sensors-194-8894, entitled Data Fusion Methods in Indoor Positioning Systems Based on Channel State Information Fingerprinting

We would like to express our gratitude to the editors and the anonymous reviewers for their constructive suggestions and criticism. The comments are well taken, and the manuscript has been revised accordingly. Below please find our responses to the reviewers’ comments. Also, for the reviewers’ convenience, major changes are written in YELLOW in the revised manuscript.

Responses to the Comments of Reviewer 3

We thank the Reviewer for the comments, suggestions and questions that helped us to improve the quality of the manuscript.

The topic of this paper is interesting and up to date, the manuscript has a practical character. Overall, it is informative and written in proper English. This is a long manuscript (30+ pages), the number of cited references is very large. However, at first glance, there seems to be an issue with the formatting and editorial side. Additionally, symbols and mathematical equations are not properly and uniformly formatted.

The experimental part is scientifically sound, the number of utilized equipment (e.g., 225 and 110 RPs) is very nice.

Suggestions and comments:

Comment #1: Check how to properly prepare the Authors and Affiliation section and make necessary corrections.

Response #1: Thank you for your comment and in the revised version we have corrected it accordingly.

Comment #2: The title of the paper should have [based on] in its name.

 Response #2: Thank you for your comment and in the revised version we have corrected it as:Data Fusion Methods in Indoor Positioning Systems Based on Channel State Information Fingerprinting

Comment #3: Numerous mathematical symbols and equations seem oddly stretched, they are not properly formatted.

 Response #3: Thank you for your comment and in the revised version we have properly formatted it.

Comment #4: Additionally, check the spacing between subsequent lines, utilized fonts, etc.

 Response #4: Thank you for your suggestion and in the revised version we have checked all the formatting requirements as per indicated.

Comment #5: Figures and block diagrams could be inserted in higher resolutions. Currently, many of them do not have sharp edges, they look blurred. They also lack proportion.

 Response #5: Thank you for your suggestion and in the revised version we have inserted all figures and block diagrams in higher resolutions.

Comment #6: They way that Figure 4 was modified (labelling the axes) shows how not to do it. Those need to be reedited once again.

Response #6: Thank you for your comment. We have corrected it as follows:

Fig. 4 Distribution of the Reference Points for generating CSI based Fingerprints during September and October, 2020

Comment #7: Other latter figures do look different. All plots/charts should be uniformly formatted. Check the style, fonts, their size, etc.

 Response #7: Thank you for your comments. We have corrected accordingly.

Comment #8:  After going through the whole paper, I am not sure about the research campaign. Authors should highlight how many devices did they utilize. What kind of devices were available? What was their technical specification? Why did you select them? Why did you perform your experiments in such an environment, etc.

Response #8: Thank you for the questions. Section 4 has mainly addressed the concerns and the presents a number of real-world experiments that were carried out on various occasions or on separate days to measure the temporal signal fluctuations using the fingerprints of the channel state information (interested in variations among the measurement days), and we assess our proposed algorithms as they were applied to these CSI real measurements of various in distributions. First, experimental conditions, datasets, and an analysis of the algorithms' overall performance were presented. Moreover, the experiments were carried out at Huawei on an area of 75m2 which contained 225 and 110 reference points (RPs) respectively, all of which were evenly dispersed (>=0.5m) from each other as shown in Figure 3. A first study was conducted in September 2020, and a second study was conducted in October 2020. For the first month, eight measurements were taken on eight different days. In contrast, five different measurements were taken on five separate days in the second month. To create the entire CSI fingerprint database, four base stations and one transmitter were available to collect channel state information from a location server. And the number of reference points collected per day is different in both months. And the total number of reference points in each month are also different. Since the measurements under study are naturally unbalanced, our analysis must account for possible discrepancies caused by the imbalance of data. Details of the system description used in the study are provided in Table 2 below and the layout and environmental settings of the real-life experimental scenarios that generate for the various datasets are depicted in Figures 3 and 4 below.

Table 2: System description

Spectrum

5.8GHz

Bandwidth

100MHz

Subcarrier bandwidth

120kHz

Label position frequency

1Hz

Comment #9: Furthermore, the References section is not properly and uniformly formatted.

 Response #9: Thank you for your comments. We have corrected accordingly.

Comment #10: The Conclusions part is far too short. Do provide additional feedback on your findings for the potential reader. Do mention about open issues and future study directions that might inspire other researchers. Comments are necessary.

 Response #10: Thank you for your suggestions. We have rewritten the conclusions section accordingly. We have also suggested future research directions based on our findings as follows:

In this paper, we considered various scenarios of temporal variations that generated the CSI-based fingerprinting measurements applied to indoor environment settings aimed at Vehicles' indoor parking lots. Along with this, extensive real-life experiments were conducted at Huawei company in a different time with an area of 75m2 constituting 225 and 110 reference points (RPs) in total collected over separated dates on September and October 2020 respectively and each RP was equidistant (>=0.5m) from the adjust next reference point. The number of measurements considered from each antenna of a base station was unequal in size and similarly, the number of labels was not totally balanced, and hence our analysis was used the feature scaling technique to avoid possible discrepancies created due to the imbalance of data. To this end, we proposed a positive knowledge transfer-based heterogeneous data fusion method for representing the different scenarios of temporal variations of CSI-based fingerprint measurements generated in a complex indoor environment targeting indoor parking lots while reducing training calibration overhead. Extensive experiments were carried out with real-world scenarios of the indoor parking phenomenon, and experimental results revealed that the proposed algorithm proved to be an efficient algorithm with consistent positioning accuracy across all possible variations. The proposed algorithm not only improves indoor parking location accuracy, but it is also a computationally robust and efficient estimator for the dynamic indoor environment unlike the decision tree and random forest algorithms significantly affected by the temporal signal fluctuations.

Results also show that the training computational complexity was much higher for all algorithms than testing time in general and exceptionally too much computational complexity was demanded by the proposed algorithm for both training and testing phases and followed by the support vector machine and neural network algorithms. Furthermore, an interesting finding was observed that the vehicle’s indoor parking lots of CSI-based fingerprints of fused data based PCA method and the indoor parking positioning performance significantly improves after transfer learning method was applied for both the fused data and the separate datasets. However, the positioning performance for the separate datasets of both months has significantly reduced after principal component was used amid to minimize computational cost. This revealed that the use of PCA method or considering 95% of the total variations of the model could not represent the entire dynamics of the environment but knowledge transfer could leverage from the training instances to enhance target positioning. In contrast, the positioning performance for the fused data of month 2, October 2020 has maintained equal performance as that of the performances achieved in separate dataset while significant improvement was observed in computational cost. Moreover, heterogeneous knowledge transfer of the fused data based PCA not only improved indoor parking positioning performance but was also significantly efficient in terms of computational cost as depicted in Figure 8 and 9. This exactly coincides with the authors claim that the fused data is significantly important for representing the signal fluctuations-based CSI-fingerprints in a dynamic environment typically in an underground environment.

The CRLB analysis technique was also applied to estimate the lower bound of the position error variance aimed at indoor parking lots. Similarly, different scenarios of temporal variations were also considered for CRLB analysis of CSI-based fingerprint measurements applied to indoor environment settings such as vehicles’ indoor parking lots. Thus, the analytical derivation of the CRLB analysis revealed that the lower bound of the variance of the location estimator depends on a) the angle of the base stations b) the number of base stations c) the distance between the target and base station,  d) correlation of features, and e) the signal propagation parameters  and . Moreover, experimental results have shown that the number of antennas of a base station could affect the lower bound location estimation error by generating a higher dimension of features unless the most significant predictors are selected otherwise the accuracy of positioning performance could be degraded due to the dimensionality curse. This analytical derivation also revealed that the fused data have shown the hybrid effect of the temporal signal variations that could come through time differences when measurements were being taken.

The database consists several values of different features as clearly mentioned on section 4.2 and the CSI fingerprint measurements features collected during the survey time were complete information that assisted in determining target positioning, including CSI real measurements, CSI imaginary measurements, latitude, longitude, coordinate systems, time of arrival, angle of arrival, and other relevant data for all scenarios. Thus, the original database is a vast one consisting various information. But, we extract a database from a Database that suit to our research goal. Following are the recommendations we forwarded as future research directions:

(1) Despite the fact that we limited our scope to CSI-amplitude information based on our objective, the phase information of CSI can also be used as fingerprints to location.

(2) Fusion of various signal measurements also could result in a robust and efficient estimator for parking lots although advanced signal processing technology is required in real life to minimize computational cost.

(3) Correlation feature extraction of the various signal metrics can also be considered in addressing signal fluctuations resulted due to the dynamic environment.

(4) Effective data preprocessing approaches are highly recommended in improving the positioning performance.

(5) Handling the data dimensionality curse could also improve the computational complexity. Thus, various approaches of data reduction techniques are highly important to be applied.

Comment #11: To sum up, this is a good paper. However, Authors did not pay much attention when preparing their manuscript. Its current form lacks adequate and precise formatting.

Response #11: Thank you for your suggestions. In the revised version, we have addressed all the concerns raised.

Round 2

Reviewer 2 Report

The paper proposes a novel algorithm for indoor positioning based on CSI and transfer learning. 

The key point is to consider transfer learning for a better positionning accuracy using  data measurements  from various days and previous model training.

I'd suggest to wait until ref 37 be published and available or add an annex to detail the transfer learning algorithm. Without it the readers don't have enough background to understand and reproduce if needed the proposed algorithm.

Some other comments or typos : 

line 153 : is -> are essential

line 234 : the AWGN is a vector, so covariance identity matrix but not a scalar sigma

line 234 : sigma is also environment factor, may be confusing notation, also used in eq.30

line 246 : not k but j=sqrt(-1) in the exponent

line 254 : |Hm| or  |H|m ?

line 278 : dB not dBm, and signal gain (not signal strength)

theta_m is not the same as in eq.10 and eq.11

line 349 : does csi amplitude refers to a 'csi eff' amplitude as in eq.6 ?

Each CSI has L streams, what represents "a", is it for Tx, antenna, Rx antenna ?

Define better na and nc, and how they are  related to nR, nT, m. 

I suppose nc is the number of CSI measurements at each RP, is this correct ? 

line 448 to 456 : unclear notations i', n', n't, j', is it n or n' ?

line 465 : algorithm 2, I don't understand the convergence criteria, could you detail or give it in annex ?

line 466 : how do  you apply the estimated transfer coefficient to CSI data or source space ? 

Author Response

Reviewer #2

Notes on Revision

Re: Manuscript sensors-194-8894, entitled Data Fusion Methods in Indoor Positioning Systems Based on Channel State Information Fingerprinting

We would like to express our gratitude to the editors and the anonymous reviewers for their constructive suggestions and criticism. The comments are well taken, and the manuscript has been revised accordingly. Below please find our responses to the reviewers’ comments. Also, for the reviewers’ convenience, major changes are written in YELLOW in the revised manuscript.

Responses to the Comments of Reviewer 2

We thank the Reviewer for the comments that helped us to improve the quality of the manuscript.

Comments and Suggestions for Authors

The paper proposes a novel algorithm for indoor positioning based on CSI and transfer learning. 

The key point is to consider transfer learning for a better positionning accuracy using  data measurements  from various days and previous model training.

Comment #1:

I'd suggest to wait until ref 37 be published and available or add an annex to detail the transfer learning algorithm. Without it the readers don't have enough background to understand and reproduce if needed the proposed algorithm.

Response #1: Thank you for your valuable suggestion.” We have only used ref 37 to describe our hypothesis on the signal fluctuations of the RSS measurements. Thus, the results of ref [37], is used to strengthen our hypothesis such that:

“The signal fluctuations of the RSS measurements of both instances of training and testing datasets for Wi-Fi APs of AP 1, AP 2, AP 3, AP 4, and AP 5 were reported to have standard deviations (in dB) of 16.8, 15.9, 14.5, 17.9, and 17.1 and 15.63, 15.14, 14.40, 17.92, and 0.00, respectively, as illustrated in Tables 1 and 2 of [37]. This experimental result [37] confirms that the temporal variations in signal distributions have a significant impact on indoor positioning performance-based RSS fingerprints, owing to the effects of multipath, NLOS, and channel conditions such as fading, shadowing, and scattering.” However, the detailed work of this current manuscript of Data Fusion Methods in Indoor Positioning Systems Based on Channel State Information Fingerprinting” is already discussed within the text document itself and the approaches are also different. They are two independent works.

At the same time, we also concur with the idea to wait until reference 37 is published as it would be better to cite a published material rather than mentioning as unpublished work in a reference list.

Some other comments or typos:

Comment #1: line 153 : is -> are essential

Response #1: Thank you for your valuable suggestion.” In the revised version, we have corrected it.

Comment #2: line 234: the AWGN is a vector, so covariance identity matrix but not a scalar sigma

, where is an identity matrix.

Response #2: Thank you for your valuable suggestion. In the revised version, we have corrected it as:

the AGWN (additive Gaussian white noise) respectively such that . Where  is an identity matrix.

Comment #3: line 234: sigma is also environment factor, may be confusing notation, also used in eq.30

Response #3: Thank you for your valuable suggestion. In the revised version, we have used  for the environmental factor.

Comment #4: line 246: not k but j=sqrt(-1) in the exponent

Response #4: Thank you for your valuable suggestion. We have corrected it as:

Comment #5: line 254: |Hm| or  |H|m ?

Response #5: Thank you for your valuable suggestion. We have corrected it to  |Hm|.

Comment #6: line 278 : dB not dBm, and signal gain (not signal strength)

theta_m is not the same as in eq.10 and eq.11

Response #6: Thank you for your valuable suggestion. We have corrected them and in this =  since we used m as index for the subcarrier, we are referring to the quantity.

Comment #7: line 349 : does csi amplitude refers to a 'csi eff' amplitude as in eq.6 ?

Each CSI has L streams, what represents "a", is it for Tx, antenna, Rx antenna ?

Define better na and nc, and how they are  related to nR, nT, m. 

I suppose nc is the number of CSI measurements at each RP, is this correct ? 

Response #7: Thank you for your valuable suggestion. Yes, csi eff amplitude is the filtered csi amplitude values.

  • Suppose denote the  CSI amplitude value at the  RP of the  BS from receiver antenna (Rx). The   refers the time when the data were collected specifically measured in number of days.  and  are the number of CSIs collected from an antenna of a BS (the number of CSI measurements at each RP) and number of antennas of a BS respectively.
  • I suppose nc is the number of CSI measurements at each RP, is this correct ? Of course, yes.

Comment #8: line 448 to 456: unclear notations i', n', n't, j', is it n or n' ?

Response #8: Thank you for your valuable suggestion. The quote is used to identify that it is pre-processed measurements or to differentiate from the original measurements. The objective function is minimized over the derived new feature spaces.

Comment #9: line 465: algorithm 2, I don't understand the convergence criteria, could you detail or give it in annex?

Response #9: Thank you for your valuable suggestion. In the revised version, we have omitted it. Our point was, the convergence criteria for instance the max. iteration required to run all the algorithms can matter to some extent.

Comment #10: line 466: how do  you apply the estimated transfer coefficient to CSI data or source space ? 

Response #10: Thank you for your valuable suggestion.

The transfer coefficients  constraint is to minimize the amplitude measurements of fluctuations between the instances of both training and testing dataset. The objective function's equality constraint would assign higher weights to the most related source instances and lower weights to the least related source instances. The new feature vectors were used to minimize the variations of the amplitude values of CSI based fingerprints and the transfer coefficients can be estimated as:

                                  (24)

We have used Lagrangian multiplier method to solve the constrained optimization problem of equation (24) and we assumed the location estimate at the iteration is obtained (mapped into 2D), and we need to estimate the location of the actual target at the iteration denoted by . CSI fingerprints are collected from each Wi-Fi access point at multiple locations within the reference points (RPs) defined during the training phase. A predictive model is trained to characterize the signal-to-location learning relationship. During the testing phase, the learned model is then used to predict the target's location based on the new received CSI measurement.

Reviewer 3 Report

Thank you for referring to my suggestions and comments. The revision has made the paper even better. Therefore, I do recommend it to be accepted and published in the Journal. Surely, I will be one of the first readers to acquaint with it online.

The editorial and formatting side would have to be adjusted at the next step, along with the References section.

Author Response

Reviewer #3-rd2:

Notes on Revision

Re: Manuscript sensors-194-8894, entitled Data Fusion Methods in Indoor Positioning Systems Based on Channel State Information Fingerprinting

We would like to express our gratitude to the editors and the anonymous reviewers for their constructive suggestions and criticism. The comments are well taken, and the manuscript has been revised accordingly. Below please find our responses to the reviewers’ comments. Also, for the reviewers’ convenience, major changes are written in YELLOW in the revised manuscript.

Responses to the Comments of Reviewer 3

We thank the Reviewer for the comments, suggestions and questions that helped us to improve the quality of the manuscript.

Comments and Suggestions for Authors

Thank you for referring to my suggestions and comments. The revision has made the paper even better. Therefore, I do recommend it to be accepted and published in the Journal. Surely, I will be one of the first readers to acquaint with it online.

We appreciate all of your comments and ideas, which helped to significantly raise the paper's quality. We also consider your review to be a valuable career lesson. Thank you so much for your constructive suggestions and comments!

Comment #1: The editorial and formatting side would have to be adjusted at the next step, along with the References section.

Response #1: Thank you for your comment and in the revised version we have corrected it accordingly based on APA style as:

References

  • Elhousni, M., & Huang, X. (2020, October). A survey on 3d lidar localization for autonomous vehicles. In 2020 IEEE Intelligent Vehicles Symposium (IV)(pp. 1879-1884). IEEE.
  • Pecoraro, G., Di Domenico, S., Cianca, E., & De Sanctis, M. (2018). CSI-based fingerprinting for indoor localization using LTE signals. EURASIP Journal on Advances in Signal Processing2018(1), 1-18.
  • Geiger, A., Lenz, P., & Urtasun, R. (2012, June). Are we ready for autonomous driving? the kitti vision benchmark suite. In 2012 IEEE conference on computer vision and pattern recognition(pp. 3354-3361). IEEE.
  • Spiekermann, S. (2004). General Aspects of Location-Based Services. In J. Schiller, & A. Voisard, Location-Based Services (pp. 9-26). Elsevier.
  • Mirama, V. F., Diez, L. E., Bahillo, A., & Quintero, V. (2021). A Survey of Machine Learning in Pedestrian Localization Systems: Applications, Open Issues and Challenges. IEEE Access9, 120138-120157.
  • Sithole, G., & Zlatanova, S. (2016). Position, location, place and area: An indoor perspective. ISPRS Ann. Photogramm. Remote Sens. Spat. Inf. Sci3(4), 89-96.
  • Buczkowski, A. Location-Based Services—Applications. 2011. Available online: https://geoawesomeness.com/knowledge-base/ location-based-services/location-based-services-applications/ (accessed on 28 December 2021).
  • Guo, X., Ansari, N., Li, L., & Duan, L. (2020). A hybrid positioning system for location-based services: Design and implementation. IEEE Communications Magazine58(5), 90-96.
  • Enge, P., & Misra, P. Special issue on GPS: The global positioning system, 1999. Proceedings of the IEEE3, 172.
  • Bulusu, N., Heidemann, J., & Estrin, D. (2000). GPS-less low-cost outdoor localization for very small devices. IEEE personal communications7(5), 28-34.
  • Barter, P. (2013). Cars are parked 95% of the time. Let’s check. [Online]. Available: http://www.reinventingparking.org/2013/02/carsare-parked-95-of-time-lets-check.html. [Accessed: 24-Nov-2016].
  • Bahl, P., & Padmanabhan, V. N. (2000, March). RADAR: An in-building RF-based user location and tracking system. In Proceedings IEEE INFOCOM 2000. Conference on computer communications. Nineteenth annual joint conference of the IEEE computer and communications societies (Cat. No. 00CH37064)(Vol. 2, pp. 775-784). Ieee.
  • Wu, K., Xiao, J., Yi, Y., Chen, D., Luo, X., & Ni, L. M. (2012). CSI-based indoor localization. IEEE Transactions on Parallel and Distributed Systems24(7), 1300-1309.
  • Reddy, H., Chandra, M. G., Balamuralidhar, P., Harihara, S. G., Bhattacharya, K., & Joseph, E. (2007, January). An improved time-of-arrival estimation for WLAN-based local positioning. In 2007 2nd International Conference on Communication Systems Software and Middleware(pp. 1-5). IEEE.
  • Gentner, C., & Jost, T. (2013, October). Indoor positioning using time difference of arrival between multipath components. In International conference on indoor positioning and indoor navigation(pp. 1-10). IEEE.
  • Tay, B., Liu, W., & Zhang, D. H. (2009, June). Indoor angle of arrival positioning using biased estimation. In 2009 7th IEEE International Conference on Industrial Informatics(pp. 458-463). IEEE.
  • Ahriz, I., Oussar, Y., Denby, B., & Dreyfus, G. (2010, March). Carrier relevance study for indoor localization using GSM. In 2010 7th Workshop on Positioning, Navigation and Communication(pp. 168-173). IEEE.
  • Roos, T., Myllymäki, P., Tirri, H., Misikangas, P., & Sievänen, J. (2002). A probabilistic approach to WLAN user location estimation. International Journal of Wireless Information Networks9(3), 155-164.
  • Chang, K. H. (2014). Bluetooth: a viable solution for IoT?[Industry Perspectives]. IEEE Wireless Communications21(6), 6-7.
  • Wang, C. S., Huang, C. H., Chen, Y. S., & Zheng, L. J. (2009, December). An implementation of positioning system in indoor environment based on active RFID. In 2009 Joint Conferences on Pervasive Computing (JCPC)(pp. 71-76).
  • Alarifi, A., Al-Salman, A., Alsaleh, M., Alnafessah, A., Al-Hadhrami, S., Al-Ammar, M. A., & Al-Khalifa, H. S. (2016). Ultra wideband indoor positioning technologies: Analysis and recent advances. Sensors16(5), 707.
  • Hu, X., Cheng, L., & Zhang, G. (2011, December). A Zigbee-based localization algorithm for indoor environments. In Proceedings of 2011 International Conference on Computer Science and Network Technology(Vol. 3, pp. 1776-1781).
  • Yasir, M., Ho, S. W., & Vellambi, B. N. (2014). Indoor positioning system using visible light and accelerometer. Journal of Lightwave Technology32(19), 3306-3316.
  • Shu, Y., Bo, C., Shen, G., Zhao, C., Li, L., & Zhao, F. (2015). Magicol: Indoor localization using pervasive magnetic field and opportunistic WiFi sensing. IEEE Journal on Selected Areas in Communications33(7), 1443-1457.
  • Karunatilaka, D., Zafar, F., Kalavally, V., & Parthiban, R. (2015). LED based indoor visible light communications: State of the art. IEEE communications surveys & tutorials17(3), 1649-1678.
  • Mainetti, L., Palano, L., Patrono, L., Stefanizzi, M. L., & Vergallo, R. (2014, September). Integration of RFID and WSN technologies in a Smart Parking System. In 2014 22nd international conference on software, telecommunications and computer networks (SoftCOM)(pp. 104-110). IEEE.
  • Tsiropoulou, E. E., Baras, J. S., Papavassiliou, S., & Sinha, S. (2017). RFID-based smart parking management system. Cyber-Physical Systems3(1-4), 22-41.
  • Gao, Y., Liu, S., Atia, M. M., & Noureldin, A. (2015). INS/GPS/LiDAR integrated navigation system for urban and indoor environments using hybrid scan matching algorithm. Sensors15(9), 23286-23302.
  • Kim, S. T., Fan, M., Jung, S. W., & Ko, S. J. (2020). External Vehicle Positioning System Using Multiple Fish-Eye Surveillance Cameras for Indoor Parking Lots. IEEE Systems Journal15(4), 5107-5118.
  • Wolcott, R. W., & Eustice, R. M. (2014, September). Visual localization within lidar maps for automated urban driving. In 2014 IEEE/RSJ International Conference on Intelligent Robots and Systems(pp. 176-183). IEEE.
  • Ichihashi, H., Notsu, A., Honda, K., Katada, T., & Fujiyoshi, M. (2009, August). Vacant parking space detector for outdoor parking lot by using surveillance camera and FCM classifier. In 2009 IEEE International Conference on Fuzzy Systems(pp. 127-134). IEEE.
  • Mackey, A., Spachos, P., & Plataniotis, K. N. (2020). Smart parking system based on bluetooth low energy beacons with particle filtering. IEEE Systems Journal14(3), 3371-3382.
  • Lin, C. H., Chen, L. H., Wu, H. K., Jin, M. H., Chen, G. H., Gomez, J. L. G., & Chou, C. F. (2019). An indoor positioning algorithm based on fingerprint and mobility prediction in RSS fluctuation-prone WLANs. IEEE Transactions on Systems, Man, and Cybernetics: Systems51(5), 2926-2936.
  • Costilla-Reyes, O., & Namuduri, K. (2014, October). Dynamic Wi-Fi fingerprinting indoor positioning system. In 2014 International Conference on Indoor Positioning and Indoor Navigation (IPIN)(pp. 271-280). IEEE.
  • Qin, S., & Guo, X. (2021). Robust Source Positioning Method With Accurate and Simplified Worst-Case Approximation. IEEE Transactions on Vehicular Technology71(2), 1891-1900.
  • Dinh-Van, N., Nashashibi, F., Thanh-Huong, N., & Castelli, E. (2017, March). Indoor Intelligent Vehicle localization using WiFi received signal strength indicator. In 2017 IEEE MTT-S international conference on microwaves for intelligent mobility (ICMIM)(pp. 33-36). IEEE.
  • Guo, X., Gidey, H.T., Zhong, K., L., Lin, L., Zhang, Y., Huang, Y. Online Transfer Learning Theories and Methods for Fingerprint-based Indoor Positioning. in press.
  • Gidey, H. T., Guo, X., Li, L., & Zhang, Y. (2022). Heterogeneous Transfer Learning for Wi-Fi Indoor Positioning Based Hybrid Feature Selection. Sensors22(15), 5840. https://doi.org/10.3390/s22155840
  • Wang, X., Gao, L., Mao, S., & Pandey, S. (2016). CSI-based fingerprinting for indoor localization: A deep learning approach. IEEE Transactions on Vehicular Technology66(1), 763-776.
  • Shi, Z., Wei, L., & Xu, Y. (2021, October). CSI-based Fingerprinting for Indoor Localization with Multi-scale Convolutional Neural Network. In 2021 IEEE 3rd Eurasia Conference on IOT, Communication and Engineering (ECICE)(pp. 233-237). IEEE.
  • Li, L., Guo, X., Zhang, Y., Ansari, N., & Li, H. (2022). Long Short-Term Indoor Positioning System via Evolving Knowledge Transfer. IEEE Transactions on Wireless Communications.
  • Yang, Z., Zhou, Z., & Liu, Y. (2013). From RSSI to CSI: Indoor localization via channel response. ACM Computing Surveys (CSUR)46(2), 1-32.
  • Chapre, Y., Ignjatovic, A., Seneviratne, A., & Jha, S. (2014, September). Csi-mimo: Indoor wi-fi fingerprinting system. In 39th annual IEEE conference on local computer networks(pp. 202-209). IEEE.
  • Ferrand, P., Decurninge, A., & Guillaud, M. (2020, December). DNN-based localization from channel estimates: Feature design and experimental results. In GLOBECOM 2020-2020 IEEE Global Communications Conference(pp. 1-6). IEEE.
  • Nessa, A., Adhikari, B., Hussain, F., & Fernando, X. N. (2020). A survey of machine learning for indoor positioning. IEEE access8, 214945-214965.
  • Regani, S. D., Xu, Q., Wang, B., Wu, M., & Liu, K. R. (2019). Driver authentication for smart car using wireless sensing. IEEE Internet of Things Journal7(3), 2235-2246.
  • Xiao, J., Wu, K., Yi, Y., & Ni, L. M. (2012, July). FIFS: Fine-grained indoor fingerprinting system. In 2012 21st international conference on computer communications and networks (ICCCN)(pp. 1-7). IEEE.
  • Wu, Z. L., Li, C. H., Ng, J. K. Y., & Leung, K. R. (2007). Location estimation via support vector regression. IEEE Transactions on mobile computing6(3), 311-321.
  • Wang, Y., Xiu, C., Zhang, X., & Yang, D. (2018). WiFi indoor localization with CSI fingerprinting-based random forest. Sensors18(9), 2869.
  • Tran, Q., Tantra, J. W., Foh, C. H., Tan, A. H., Yow, K. C., & Qiu, D. (2006, September). Wireless indoor positioning system with enhanced nearest neighbors in signal space algorithm. In IEEE Vehicular Technology Conference(pp. 1-5). IEEE.
  • Sobehy, A., Renault, E., & Mühlethaler, P. (2020, June). CSI-MIMO: K-nearest neighbor applied to indoor localization. In ICC 2020-2020 IEEE International Conference on Communications (ICC)(pp. 1-6). IEEE.
  • Wu, Z., Jiang, L., Jiang, Z., Chen, B., Liu, K., Xuan, Q., & Xiang, Y. (2018). Accurate indoor localization based on CSI and visibility graph. Sensors18(8), 2549.
  • Yong, Z., & Chengbin, W. (2021, March). An indoor positioning system using Channel State Information based on TrAdaBoost Tranfer Learning. In 2021 4th International Conference on Advanced Electronic Materials, Computers and Software Engineering (AEMCSE)(pp. 1286-1293). IEEE.
  • Han, D., Jung, S., Lee, M., & Yoon, G. (2014). Building a practical Wi-Fi-based indoor navigation system. IEEE Pervasive Computing13(2), 72-79.
  • Qin, S., & Guo, X. (2022). IoT edge computing-enabled efficient localization via robust optimal estimation. IEEE Internet of Things Journal.
  • Dardari, D., Chong, C. C., & Win, M. (2008). Threshold-based time-of-arrival estimators in UWB dense multipath channels. IEEE Transactions on Communications56(8), 1366-1378.
  • He, S., & Chan, S. H. G. (2015). Wi-Fi fingerprint-based indoor positioning: Recent advances and comparisons. IEEE Communications Surveys & Tutorials18(1), 466-490.
  • Chang, D. C., & Fan, M. W. (2014, August). Aoa target tracking with new imm pf algorithm. In 2014 IEEE 57th International Midwest Symposium on Circuits and Systems (MWSCAS)(pp. 729-732). IEEE.
  • Wang, B., Zhou, S., Liu, W., & Mo, Y. (2014). Indoor localization based on curve fitting and location search using received signal strength. IEEE transactions on industrial electronics62(1), 572-582.
  • Xiao, J., Zhou, Z., Yi, Y., & Ni, L. M. (2016). A survey on wireless indoor localization from the device perspective. ACM Computing Surveys (CSUR)49(2), 1-31.
  • Shahmansoori, A., Garcia, G. E., Destino, G., Seco-Granados, G., & Wymeersch, H. (2017). Position and orientation estimation through millimeter-wave MIMO in 5G systems. IEEE Transactions on Wireless Communications17(3), 1822-1835.
  • Yin, J., Wan, Q., Yang, S., & Ho, K. C. (2015). A simple and accurate TDOA-AOA localization method using two stations. IEEE Signal Processing Letters23(1), 144-148.
  • Sharp, I., & Yu, K. G. (2014). Indoor TOA error measurement, modeling, and analysis. IEEE Transactions on Instrumentation and Measurement, 63(9), 2129–2144
  • Alavi, B., & Pahlavan, K. (2006). Modeling of the TOA-based distance measurement error using UWB indoor radio measurements. IEEE Communications Letters, 10(4), 275–277.
  • Hernandez, A., Badorrey, R., Choliz, J., Alastruey, I., & Valdovinos, A. (2008). Accurate indoor wireless location with IR UWB systems a performance evaluation of joint receiver structures and TOA based mechanism. IEEE Transactions on Consumer Electronics, 54(2), 381–389.
  • Wen, F., & Liang, C. (2014). Fine-grained indoor localization using single access point with multiple antennas. IEEE Sensors Journal15(3), 1538-1544.
  • Taponecco, L., D’Amico, A. A., & Mengali, U. (2011). Joint TOA and AOA estimation for UWB localization applications. IEEE Transactions on Wireless Communications, 10(7), 2207–2217.
  • Liu, H., Darabi, H., Banerjee, P., & Liu, J. (2007). Survey of wireless indoor positioning techniques and systems. IEEE Transactions on Systems, Man, Cybernetics, Systems Part C—Applications and Reviews, 37(6), 1067–1080.
  • Basri, C., & El Khadimi, A. (2016, September). Survey on indoor localization system and recent advances of WIFI fingerprinting technique. In 2016 5th international conference on multimedia computing and systems (ICMCS)(pp. 253-259). IEEE.
  • Liu, W., Cheng, Q., Deng, Z., Chen, H., Fu, X., Zheng, X., ... & Wang, S. (2019, September). Survey on CSI-based indoor positioning systems and recent advances. In 2019 International Conference on Indoor Positioning and Indoor Navigation (IPIN)(pp. 1-8). IEEE.
  • De Bast, S., Guevara, A. P., & Pollin, S. (2020, May). CSI-based positioning in massive MIMO systems using convolutional neural networks. In 2020 IEEE 91st Vehicular Technology Conference (VTC2020-Spring)(pp. 1-5). IEEE.
  • Feng, C., Arshad, S., Yu, R., & Liu, Y. (2018, May). Evaluation and improvement of activity detection systems with recurrent neural network. In 2018 IEEE International Conference on Communications (ICC)(pp. 1-6). IEEE.
  • Hsieh, C. H., Chen, J. Y., & Nien, B. H. (2019). Deep learning-based indoor localization using received signal strength and channel state information. IEEE access7, 33256-33267.
  • Xia, P., Zhou, S., & Giannakis, G. B. (2004). Adaptive MIMO-OFDM based on partial channel state information. IEEE Transactions on signal processing52(1), 202-213.
  • F. Molisch, Orthogonal Frequency-Division Multiplexing (OFDM), New York, NY, USA: Wiley, 2011.
  • Hara, “OFDM,” in Wireless Communication Technologies: New Multimedia Systems, 2005.
  • Biglieri, R. Calderbank, A. Constantinides, A. Goldsmith, A. Paulraj, and H. V. Poor, MIMO wireless communications. 2007.
  • Wu, K., Xiao, J., Yi, Y., Gao, M., & Ni, L. M. (2012, March). FILA: Fine-grained indoor localization. In 2012 Proceedings IEEE INFOCOM(pp. 2210-2218). IEEE.
  • Wu, Z., Xu, Q., Li, J., Fu, C., Xuan, Q., & Xiang, Y. (2017). Passive indoor localization based on csi and naive bayes classification. IEEE Transactions on Systems, Man, and Cybernetics: Systems48(9), 1566-1577.
  • F. Molisch, Wireless Communications, Wiley - IEEE Series. Wiley, 2011.
  • Guo, X., Ansari, N., Hu, F., Shao, Y., Elikplim, N. R., & Li, L. (2019). A survey on fusion-based indoor positioning. IEEE Communications Surveys & Tutorials22(1), 566-594.
  • Wang, X., Gao, L., Mao, S., & Pandey, S. (2015, March). DeepFi: Deep learning for indoor fingerprinting using channel state information. In 2015 IEEE wireless communications and networking conference (WCNC)(pp. 1666-1671). IEEE.
  • Kanaris, L., Kokkinis, A., Liotta, A., & Stavrou, S. (2017, May). Combining smart lighting and radio fingerprinting for improved indoor localization. In 2017 IEEE 14th International Conference on Networking, Sensing and Control (ICNSC)(pp. 447-452). IEEE.
  • Antevski, K., Redondi, A. E., & Pitic, R. (2016, July). A hybrid BLE and Wi-Fi localization system for the creation of study groups in smart libraries. In 2016 9th IFIP wireless and mobile networking conference (WMNC)(pp. 41-48). IEEE.
  • Wang, S., Hu, D., Sun, X., Yan, S., Huang, J., Zhen, W., & Li, Y. (2018, November). A data fusion method of indoor location based on adaptive UKF. In 2018 7th International Conference on Digital Home (ICDH)(pp. 257-263). IEEE.
  • Zhang, Z., Xie, L., Zhou, M., & Wang, Y. (2020, August). CSI-based Indoor Localization Error Bound Considering Pedestrian Motion. In 2020 IEEE/CIC International Conference on Communications in China (ICCC)(pp. 811-816). IEEE.
  • Zhao, Y., Li, X., & Xu, C. Z. (2018, August). ER-CRLB Analysis for Indoor Multi-Antenna Localization System. In 2018 IEEE/CIC International Conference on Communications in China (ICCC)(pp. 379-383). IEEE.
  • Hossain, A. M., & Soh, W. S. (2010, March). Cramer-Rao bound analysis of localization using signal strength difference as location fingerprint. In 2010 Proceedings IEEE INFOCOM(pp. 1-9). IEEE.
  • Jiang, Q., Qiu, F., Zhou, M., & Tian, Z. (2016). Benefits and impact of joint metric of AOA/RSS/TOF on indoor localization error. Applied Sciences6(10), 296.
  • Catovic, A., & Sahinoglu, Z. (2004). The Cramer-Rao bounds of hybrid TOA/RSS and TDOA/RSS location estimation schemes. IEEE Communications Letters8(10), 626-628.
  • Gui, L., Yang, M., Yu, H., Li, J., Shu, F., & Xiao, F. (2017). A Cramer–Rao lower bound of CSI-based indoor localization. IEEE Transactions on Vehicular Technology67(3), 2814-2818.
  • Wang, J., Wang, X., Peng, J., Hwang, J. G., & Park, J. G. (2021, August). Indoor Fingerprinting Localization Based on Fine-grained CSI using Principal Component Analysis. In 2021 Twelfth International Conference on Ubiquitous and Future Networks (ICUFN)(pp. 322-327). IEEE.
  • Kukolj, D., Vuckovic, M., & Pletl, S. (2011, October). Indoor location fingerprinting based on data reduction. In 2011 International Conference on Broadband and Wireless Computing, Communication and Applications (pp. 327-332). IEEE.